# RNA editing at a limited number of sites is sufficient to prevent MDA5 activation in the mouse brain

Jung In Kim[1]ʘ, Taisuke Nakahama[1]ʘ, Ryuichiro Yamasaki[1], Pedro Henrique Costa Cruz[1], Tuangtong Vongpipatana[1], Maal Inoue[1], Nao Kanou[1], Yanfang Xing[1], Hiroyuki Todo[1], Toshiharu Shibuya[1], Yuki Kato[1,2], Yukio Kawahara[1,2,3]*

1 Department of RNA Biology and Neuroscience, Graduate School of Medicine, Osaka University, Suita, Japan, 2 Integrated Frontier Research for Medical Science Division, Institute for Open and Transdisciplinary Research Initiatives (OTRI), Osaka University, Suita, Japan, 3 Genome Editing Research and Development Center, Graduate School of Medicine, Osaka University, Suita, Japan

ʘ These authors contributed equally to this work.
* ykawahara@rna.med.osaka-u.ac.jp

**Data Availability Statement:** The RNA-seq data used in this study are available through the DNA Data Bank of Japan (DDBJ) under accession number DRA010640 (link: https://ddbj.nig.ac.jp/

## Abstract

Adenosine deaminase acting on RNA 1 (ADAR1), an enzyme responsible for adenosine-to-inosine RNA editing, is composed of two isoforms: nuclear p110 and cytoplasmic p150. Deletion of *Adar1* or *Adar1 p150* genes in mice results in embryonic lethality with overexpression of interferon-stimulating genes (ISGs), caused by the aberrant recognition of unedited endogenous transcripts by melanoma differentiation-associated protein 5 (MDA5). However, among numerous RNA editing sites, how many RNA sites require editing, especially by ADAR1 p150, to avoid MDA5 activation and whether ADAR1 p110 contributes to this function remains elusive. In particular, ADAR1 p110 is abundant in the mouse brain where a subtle amount of ADAR1 p150 is expressed, whereas *ADAR1* mutations cause Aicardi–Goutières syndrome, in which the brain is one of the most affected organs accompanied by the elevated expression of ISGs. Therefore, understanding RNA editing–mediated prevention of MDA5 activation in the brain is especially important. Here, we established *Adar1 p110*–specific knockout mice, in which the upregulated expression of ISGs was not observed. This result suggests that ADAR1 p150–mediated RNA editing is enough to suppress MDA5 activation. Therefore, we further created *Adar1 p110/Adar2* double knockout mice to identify ADAR1 p150–mediated editing sites. This analysis demonstrated that although the elevated expression of ISGs was not observed, only less than 2% of editing sites were preserved in the brains of *Adar1 p110/Adar2* double knockout mice. Of note, we found that some sites were highly edited, which was comparable to those found in wild-type mice, indicating the presence of ADAR1 p150–specific sites. These data suggest that RNA editing at a very limited sites, which is mediated by a subtle amount of ADAR1 p150, is sufficient to prevents MDA5 activation, at least in the mouse brain.

DRASearch/submission?acc=DRA010640 and
https://www.ncbi.nlm.nih.gov/sra/?term=
DRA010640).

**Funding:** This work was supported by Grants-in-Aid KAKENHI (19K22580 and 20H03341 to Y. Kw., 18K15186 to T.N., 20J11266 to J.I.K., 18J11733 to T.V. and 18K11526 to Y. Ka.) - https://www.jsps.go.jp/english/e-grants/index.html, from the Ministry of Education, Culture, Sports, Science and Technology (MEXT) of Japan. A grant (JP20ek0109433 to T.N.) - https://www.amed.go.jp/en/index.html, from the Japan Agency for Medical Research and Development (AMED); and by grants from The Tokyo Biochemical Research Foundation (to Y. Kw.) / http://www.tokyobrf.or.jp/; The Naito Foundation (to T.N.) - https://www.naito-f.or.jp/en/; and Novartis Research Grants (to T.N.) -https://www.novartis.co.jp/innovation/support-activity/basic-research; The Mochida Memorial Foundation for Medical and Pharmaceutical Research (to T.N.) - https://www.mochidazaidan.or.jp; Astellas Foundation for Research on Metabolic Disorders (to T.N.) - https://www.astellas-foundation.or.jp; The Uehara Memorial Foundation (to T.N.) - https://www.ueharazaidan.or.jp/; and The Osaka Medical Research Foundation for Intractable Diseases (to T.N.) - https://nanbyo.or.jp/; and the Takeda Science Foundation (to Y. Kw. and T.N.). P.H.C.C. and Y.X. were supported by MEXT scholarships - https://www.mext.go.jp/a_menu/koutou/ryugaku/06032818.htm. J.I.K. was supported by The Korean Scholarship Foundation - http://www.korean-s-f.or.jp/. J.I.K. and T.V. were supported by a Research Fellowship for Young Scientists from the Japan Society for the Promotion of Science (JSPS). The funders had no role in study design, data collection and analysis, decision to publish, or preparation of the manuscript.

**Competing interests:** The authors have declared that no competing interests exist.

## Author summary

RNA is subject to various post-transcriptional modifications, which add the information to regulate the fate of each RNA. One such modification is RNA editing, in which certain adenosine in double-stranded RNAs is converted to inosine by deamination reaction that is catalyzed by adenosine deaminases acting on RNA (ADARs). ADAR1 and ADAR2 are active editing enzymes in mammals. In addition, ADAR1 is composed of nuclear p110 and cytoplasmic p150 isoforms. However, the difference in the targets and the function of each ADAR is not fully understood. Previous studies demonstrate that innate immunity is activated in *Adar1 p150* knockout mice, which is not observed in *Adar2* knockout mice. Here, we established *Adar1 p110* knockout mice. These mutant mice showed high levels of mortality during the early post-natal stages, whereas we demonstrated that this is caused by RNA editing–independent function of ADAR1 p110. We further generated *Adar1 p110/Adar2* double knockout mice, in which innate immunity is not activated, although more than 98% of all the editing sites are absent in the brain of these double KO mice. Collectively, ADAR1 150–mediated RNA editing at a very limited sites is sufficient to avoid activation of innate immunity, at least in the mouse brain.

## Introduction

Adenosine (A)-to-inosine (I) RNA editing is a post-transcriptional RNA modification that is estimated to occur at more than 100 million and 50 thousand sites in humans and mice, respectively [1–5]. In mammals, RNA editing occurs in double-stranded RNAs (dsRNAs), given that this conversion is catalyzed by adenosine deaminases acting on RNA (ADARs), ADAR1 and ADAR2, which have conserved dsRNA-binding domains in addition to a deaminase domain [6–8]. Long dsRNAs are formed by inverted repetitive sequences, such as short interspersed elements (SINEs), located in introns and 3' untranslated regions (UTRs) of mRNA, and thereby more than 90% of all RNA editing events occur within *Alu* repeats, the most common type of SINEs, in humans [1,9,10].

ADAR1 is expressed as two isoforms: interferon (IFN)-inducible longer ADAR1 p150, which is translated from the start codon present in exon 1A, and a constitutively expressed shorter ADAR1 p110, which harbors exon 1B (or possible alternative exon 1C) and is translated from the downstream AUG located in exon 2 [11–14]. ADAR1 p110 and ADAR2 are predominantly localized in the nucleus [15–19]. In contrast, ADAR1 p150, which contains a nuclear export signal within an isoform-specific Z-DNA/RNA binding domain α (Zα domain), is normally expressed in the cytoplasm, whereas this intracellular localization can be altered by nucleocytoplasmic shuttling, which is more active under certain conditions, such as viral infections [15–17,19]. We and another group recently reported that ADAR1 and ADAR2 are the sole enzymes responsible for A-to-I RNA editing *in vivo* by analyzing RNA editing sites in mutant mice deficient in activities of both ADAR1 and ADAR2 [20,21]. However, ADAR1 and ADAR2 are expressed at various levels in a tissue-specific manner. For instance, ADAR1 p110 and ADAR2 are especially abundant in the mouse brain where ADAR1 p150 is barely detectable [21–24]. In contrast, ADAR1 p150 is highly expressed in lymphoid organs such as the thymus and spleen of adult mice. Therefore, RNA editing is regulated by orchestration of differentially expressed ADAR1 and ADAR2 in a site- and tissue-specific manner [21]. Nevertheless, it remains unknown how ADAR1 p110 and ADAR1 p150, which might shuttle between the nucleus and cytoplasm [15,17,19], contribute to RNA editing at the same sites.

Both *Adar1* knockout (KO; *Adar1*$^{-/-}$) mice and *Adar1 p150*–specific KO (*Adar1 p150*$^{-/-}$) mice show embryonic lethality at E12.5 with an excess production of type I IFN, leading to the upregulated expression of IFN-stimulated genes (ISGs) [25–29]. Furthermore, *Adar1* knock-in (KI) mice harboring the editing-inactive E861A point mutation (*Adar1*$^{E861A/E861A}$ mice) also displays lethality at ~E13.5 [30], suggesting that ADAR1–mediated RNA editing is essential for early development and suppression of the overproduction of type I IFN. These phenotypes are different from postnatal lethality due to progressive seizures found in *Adar2* KO (*Adar2*$^{-/-}$) mice, which can be rescued by the insertion of a point mutation in the *Gria2* gene at the ADAR2-specific Q/R site, leading to the sole expression of the edited glutamate receptor subunit, GluA2 (*Adar2*$^{-/-}$ *Gria2*$^{R/R}$ mice) [31,32]. In contrast, embryonic lethality found in *Adar1*$^{-/-}$, *Adar1 p150*$^{-/-}$ and *Adar1*$^{E861A/E861A}$ mice is rescued by concurrent deletion of either *Ifih1*-encoded melanoma differentiation associated gene 5 (MDA5), a cytosolic sensor for viral dsRNA, or mitochondrial antiviral signaling protein (MAVS), an adaptor protein downstream of MDA5 [30,33,34]. Overexpression of ISGs in these mutant mice is also much ameliorated by deletion of either MDA5 or MAVS. This indicates that RNA editing, especially in 3'UTRs of mRNA, alters dsRNA structure, which is essential for preventing MDA5 sensing of endogenous dsRNAs as non-self. Indeed, *Alu* hybrids located in the 3'UTRs of mRNA are the predominant ligands for MDA5 in a human cell culture model [35]. Such lines of evidence suggest that at least ADAR1 p150 is required for preventing MDA5 activation, whereas the contribution of ADAR1 p110 remains undetermined. In particular, 3'UTRs of mRNA are most likely edited first by ADAR1 p110 in the nucleus and then might be further edited by cytoplasmic ADAR1 p150. Therefore, how ADAR1 p110 and p150 compensate each other for RNA editing at the same sites remains unclear. Given that ADAR1 p150 contains an isoform-specific Zα domain, there is another possibility that preferential targets differ between ADAR1 p110 and p150, and that ADAR1 p150–specific sites might be required to be edited to avoid MDA5 sensing. However, although overexpression of ISGs is observed in the fetal brain of *Adar1*$^{E861A/E861A}$ mice and is not completely normalized in the adult brain of *Adar1*$^{E861A/E861A}$ *Ifih1*$^{-/-}$ mice [36], ADAR1 p150 is expressed at the lowest level in the mouse brain [21,23]. Therefore, abundantly expressed ADAR1 p110–mediated RNA editing might contribute to suppressing MDA5 activation, at least in the brain, or a subtle amount of ADAR1 p150 might be sufficient to suppress it. In the latter case, it is possible that the number of editing sites necessary for escaping MDA5 sensing might be very limited, which also remains to be examined. Nevertheless, understanding RNA editing–mediated prevention of MDA5 activation in the brain is especially important, since *ADAR1* mutation can cause Aicardi–Goutières syndrome (AGS), a rare autosomal recessive disease, in which the brain is one of the most affected organs accompanied by a type I IFN signature [37,38].

In this study, we generated *Adar1 p110*–specific KO mice by deleting exon 1B, exon 1C, and the constitutive promoter. We found that the expression of ADAR1 p150 was not affected, indicating that the IFN-inducible promoter could control it. Furthermore, we observed no upregulated expression of ISGs in all the organs examined in *Adar1 p110*–specific KO and *Adar1 p110*/*Adar2* double KO mice. Of note, more than 98% of all the editing sites found in wild-type mice were absent in the brain of *Adar1 p110*/*Adar2* double KO mice. These results collectively suggest that sole expression of ADAR1 p150 can prevent MDA5 activation, leading to the overexpression of ISGs, whereas a limited number of the sites requires RNA editing by ADAR1 p150 to avoid MDA5 sensing, at least, in the mouse brain.

## Results

### ADAR1 p150 is expressed at very low levels in the mouse brain from birth

We previously reported that ADAR1 p110 and ADAR2 were highly expressed but that ADAR1 p150 was barely detectable in the adult mouse brain, although the latter was abundant in the

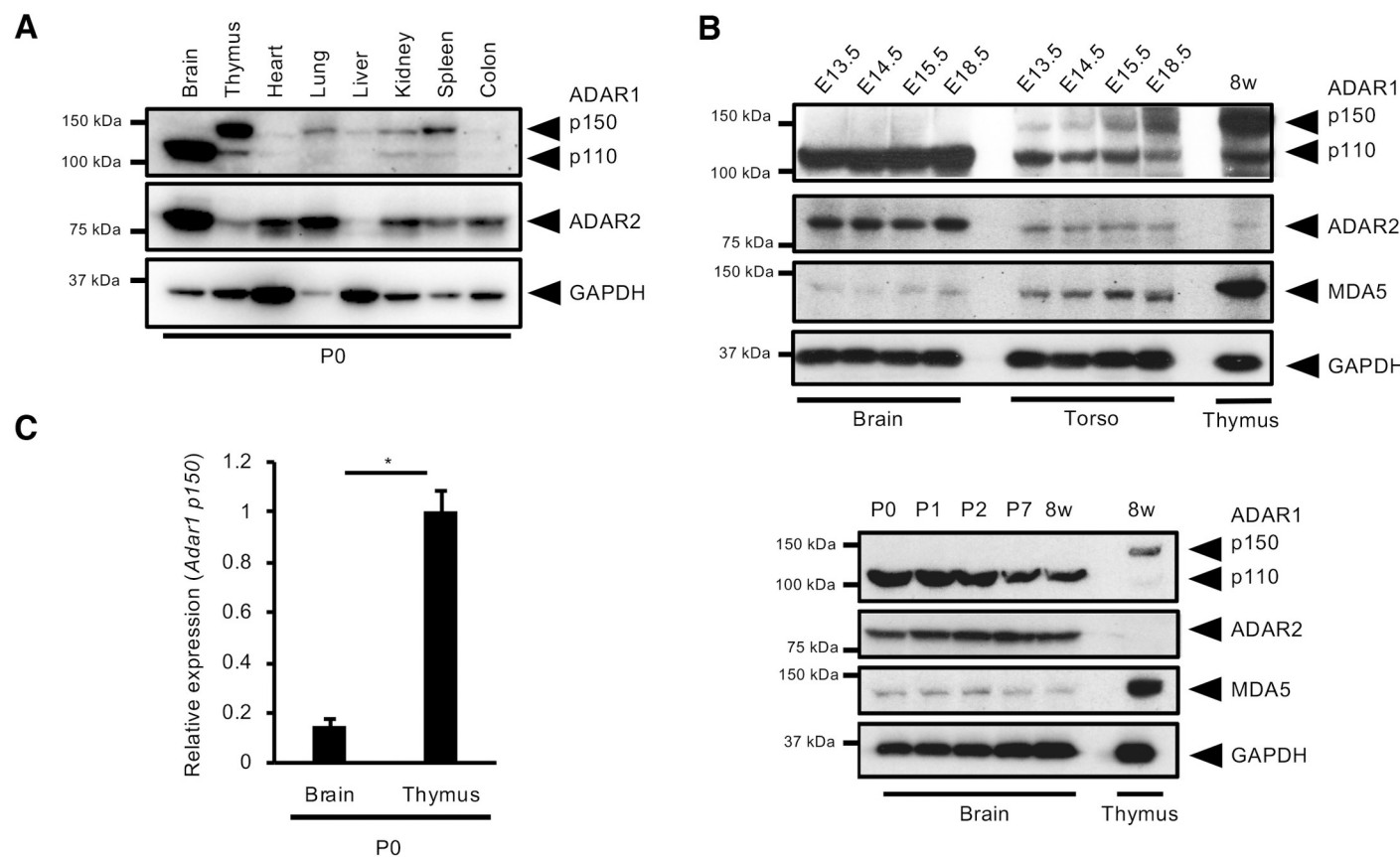

**Fig 1. Tissue-specific expression pattern of ADAR1 isoforms.** (A) Immunoblot analysis of ADAR1 p110, ADAR1 p150, and ADAR2 protein expression in various organs from wild-type mice at post-natal day 0 (P0). The expression of GAPDH protein is shown as a reference. (B) Immunoblot analysis of ADAR1 p110, ADAR1 p150, ADAR2, MDA5, and GAPDH protein expression in brains and torsos (upper panel only) isolated from wild-type mice at various embryonic stages (*upper panel*) and post-natal days (*lower panel*). The expression of these proteins in the thymus at 8 weeks is shown as a reference. (C) The relative expression of Adar1 p150 mRNA in brains and thymi from wild-type mice at P0. Values represent relative gene expression normalized to GAPDH mRNA and are displayed as the mean ± SEM (n = 4 mice for each group; Mann–Whitney *U*-test, $^*p < 0.05$).

thymus and spleen [23]. To investigate whether this expression pattern is established in the early developmental period, we first evaluated protein levels of ADAR1 and ADAR2 on post-natal day 0 (P0) and found that ADAR1 p150 was uniquely abundant in the thymus among the organs examined at P0 (**Fig 1A**). In contrast, ADAR1 p150 was barely detectable in the mouse brain. This expression pattern was established during embryonic stages and sustained until adulthood, which might have correlated with the relatively low expression of MDA5 (**Fig 1B**). Given that it is difficult to detect ADAR1 p150 by immunoblotting, we performed quantitative reverse transcription (qRT)–PCR analysis for Adar1 p150 mRNA. This indicated that the expression of Adar1 p150 mRNA in the brain was less than 20% of that in the thymus (**Fig 1C**). These results suggest that the p110 isoform is the predominant ADAR1 in the mouse brain where ADAR1 p150 is expressed at very low levels.

### *Adar1 p110*–specific KO mice reveal a high mortality rate during the early post-natal days

Given that loss of ADAR1 activates the MDA5-sensing pathway in various organs, including the mouse brain [36], it was of interest to investigate whether ADAR1 p150 contributed to preventing MDA5 activation in the mouse brain and whether ADAR1 p110 was involved in this

pathway. For this purpose, we initially intended to generate *Adar1 p110*–specific KO mice by the insertion of point mutations at the start codon of *Adar1 p110*, converting the initial methionine at residue 249 (M249) to alanine (A), which did not affect the translation of ADAR1 p150 (**S1A Fig**). *Adar1^{M249A/M249A}* mice were successfully established, whereas no overt abnormal phenotypes were observed (**S1B Fig**). Unexpectedly, immunoblotting analysis detected the expression of truncated ADAR1 p110 in the brain of *Adar1^{M249A/M249A}* mice, which was likely translated from a downstream methionine located in either exon 2 or exon 3 (**S1A and S1C Fig**). Of note, truncated ADAR1 p110 in *Adar1^{M249A/M249A}* mice was less abundant than full-length ADAR1 p110 in wild-type (*Adar1^{+/+}*) mice and was detected only in the brain among the organs examined (**S1C Fig**). This suggests that truncated ADAR1 p110 was less efficiently translated or was more unstable, leading to degradation.

Next, we took another approach to generate *Adar1 p110*–specific KO mice by deleting a ~1.6-kbp region encompassing the constitutive promoter, exon 1B and a possible alternative exon 1C, which are required for the transcription of *Adar1 p110* from the constitutive promoter (**Fig 2A**) [11,13]. We successfully obtained two lines, Lines 15 and 17, both of which had lost an almost identical genomic region and showed similar phenotypes. We therefore combined these lines for further analyses. In these mutant mice, the expression of ADAR1 p110 was selectively lost in all organs examined at P0, which did not affect the expression of ADAR1 p150 in addition to ADAR2 (**Fig 2B**). Furthermore, predominant cytoplasmic localization of ADAR1 p150 was not substantially altered in the absence of ADAR1 p110 (**S2A Fig**). In addition, the expression of ADAR1 p110 was not observed in the splenocytes of these mutant mice under the condition of the upregulated expression of ADAR1 p150, which was induced by IFN stimulation (**S2B Fig**); we thereby termed these mutant mice *Adar1 p110^{-/-}* mice. We further showed that the expression of Adar1 p150 mRNA at P0 was not altered in all organs examined (**Fig 2C**). In addition, the amount of ADAR1 p110 protein in *Adar1 p110^{+/-}* mice was less abundant than that in wild-type (*Adar1 p110^{+/+}*) mice at P0 (**Fig 2D**). These results collectively indicate that the constitutive promoter is dispensable for regulating the expression of ADAR1 p150 and that loss of ADAR1 p110 does not affect the expression and the intracellular localization of ADAR1 p150. In addition, although ADAR1 p110 may be produced from Adar1 p150 mRNA [39], it is extremely inefficient even if it does occur.

Although it is difficult to distinguish *Adar1 p110^{-/-}* pups from wild-type ones at birth, more than half of these died within two days after birth (**Fig 2E and 2F**). Surviving pups showed growth retardation during development and less than 20% of *Adar1 p110^{-/-}* mice survived after two weeks of age (**Fig 2E, 2F and 2G**). However, although this smallness is sustained throughout life, surviving *Adar1 p110^{-/-}* mice can be mature and fertile. This high mortality rate during early post-natal days is not simply attributed to competition against wild-type and heterozygous mutant pups since a similar phenomenon was observed in pups born from *Adar1 p110^{-/-}* parents. In addition, we did not detect any evident morphological abnormalities or signs of inflammation in the multiple organs examined, including the brain, heart, kidney and intestines (**S3 Fig**). Therefore, the organs responsible for the high mortality rate observed in *Adar1 p110^{-/-}* mice during early post-natal days remains undetermined. However, milk spots seemed to be small in *Adar1 p110^{-/-}* pups, which might have been caused by weak suckling or breathing. In addition, given that *Adar1^{M249A/M249A}* mice, in which truncated ADAR1 p110 is observed only in the brain, are phenotypically normal, ADAR1 p110 most likely plays a critical role in a certain region of the brain, especially during early post-natal days.

To investigate the molecular mechanism underlying the high mortality rate, especially the contribution of ADAR1 p110–mediated RNA editing, we crossed *Adar1 p110^{+/-}* mice with *Adar1^{E861A/+}* mice. The resulting *Adar1^{E861A/p110del}* mice demonstrated no obvious abnormal phenotypes, an outcome different from those in *Adar1 p110^{-/-}* mice or the embryonic lethality

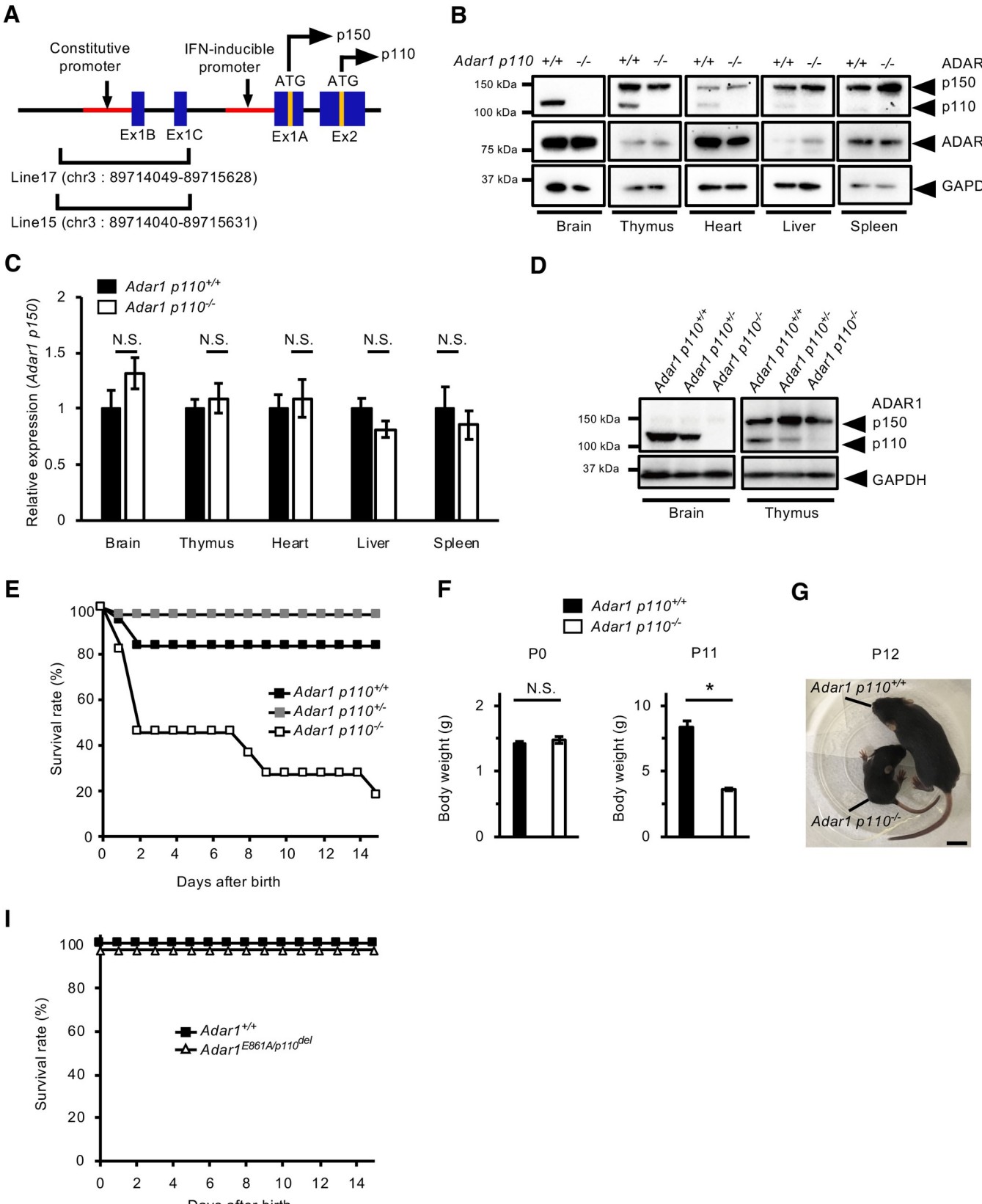

**Fig 2. Generation of *Adar1 p110*–specific KO mice. (A)** Schematic diagram of mouse *Adar1* gene, which is transcribed from either a constitutive promoter or type I interferon (IFN)-inducible promoter. Adar1 p110 transcripts usually contain exon 1B (Ex1B) or minor Ex1C and are translated from an initial methionine site (ATG) located in Ex2, while p150 transcripts contain Ex1A, which includes the initial methionine site. The genomic regions deleted in the two lines (Lines 15 and 17) of *Adar1 p110*–specific knockout (KO) mice are indicated. **(B)** The expression of ADAR1 p110, p150, and ADAR2 proteins in various organs on post-natal day 0 (P0) is compared between wild-type (*Adar1 p110$^{+/+}$*) and *Adar1 p110*–specific KO (*Adar1 p110$^{-/-}$*) mice. The expression of GAPDH protein is shown as a reference. **(C)** The relative expression of Adar1 p150 mRNA in various organs at P0 is compared between *Adar1 p110$^{+/+}$* and *Adar1 p110$^{-/-}$* mice. Values represent relative gene expression normalized to GAPDH mRNA and are displayed as the mean ± SEM (n = 3–4 mice for each group; Mann–Whitney *U*-test, N.S., not significant). **(D)** Immunoblot analysis of ADAR1 p110 and p150 protein expression in brains and thymi of *Adar1 p110$^{+/+}$*, *Adar1 p110$^{+/-}$*, and *Adar1 p110$^{-/-}$* mice at P0. The expression of GAPDH protein is shown as reference. **(E)** Survival curve of *Adar1 p110$^{+/+}$* (n = 18), *Adar1 p110$^{+/-}$* (n = 34), and *Adar1 p110$^{-/-}$* mice (n = 11). **(F)** Body weights at P0 (left panel) and P11 (right panel) are compared between *Adar1 p110$^{+/+}$* and *Adar1 p110$^{-/-}$* mice. Values are displayed as the mean ± SEM (n = 3–7 mice for each group; Mann–Whitney *U*-test, *$p < 0.05$, N.S., not significant). **(G)** Representative image of *Adar1 p110$^{+/+}$* and *Adar1 p110$^{-/-}$* mice at P12. Scale bar, 1 cm. **(I)** Survival curves of *Adar1$^{+/+}$* (n = 12) and *Adar1$^{E861A/p110del}$* mice (n = 9), which were created by crossing *Adar1 p110$^{+/-}$* mice with *Adar1$^{E861A/+}$* mice.

found in *Adar1$^{E861A/E861A}$* mice (**Fig 2I**). The result indicates that the high mortality rate found in *Adar1 p110$^{-/-}$* mice during the early post-natal period is caused by an RNA editing–independent function of ADAR1 p110.

## No activation of MDA5-sensing pathway in *Adar1 p110*–specific KO mice

To investigate the involvement of the p110 isoform in the ADAR1–mediated prevention of MDA5-sensing of endogenous dsRNAs, we first evaluated whether the expression of ISGs, which is triggered by MDA5 activation, was upregulated. This analysis detected no significant upregulation of the expression of three ISGs, *Ifih1* encoding MDA5, *Ifit1* and *Cxcl10*, in multiple organs examined including the brain in *Adar1 p110$^{-/-}$* mice (**Fig 3A**). Furthermore, concurrent deletion of MDA5 failed to restore the high mortality rate found in *Adar1 p110$^{-/-}$* mice during early post-natal days (**Fig 3B**). These results suggest that loss of ADAR1 p110 did not activate the MDA5-sensing pathway. Furthermore, to examine the possible compensation by ADAR2, we created *Adar1 p110$^{-/-}$ Adar2$^{-/-}$* and *Adar1 p110$^{-/-}$ Adar2$^{-/-}$ Gria2$^{R/R}$* mice. We observed no significant difference in terms of phenotypes between *Adar1 p110$^{-/-}$* and *Adar1 p110$^{-/-}$ Adar2$^{-/-}$ Gria2$^{R/R}$* mice. In contrast, *Adar1 p110$^{-/-}$ Adar2$^{-/-}$* mice were expected to display severe seizure, leading to lethality [31,32]. Therefore, we extracted RNAs from the brain and thymus at P0 and evaluated whether the expression of ISGs was upregulated. However, no elevated expression of ISGs was observed in *Adar1 p110$^{-/-}$ Adar2$^{-/-}$* mice (**Fig 3C**), which indicates that the sole expression of ADAR1 p150 is sufficient to suppress activation of the MDA5-sensing pathway in mouse organs, including the brain.

## Most of RNA editing sites in coding and intronic regions are absent in *Adar1 p110$^{-/-}$ Adar2$^{-/-}$* mice

To identify RNA editing sites required for preventing MDA5 activation, we compared the RNA editing pattern in the brains and thymi of *Adar1 p110$^{-/-}$* and *Adar1 p110$^{-/-}$ Adar2$^{-/-}$* mice at P0 to that of wild-type (*Adar1 p110$^{+/+}$*) mice. By analyzing more than 49 million reads per each sample, we identified more than 8,500 and 2,100 sites in the brain and thymus, respectively, of *Adar1 p110$^{+/+}$* mice, whereas more than 98% and 69% of these sites, respectively, were not detected in the corresponding organs of *Adar1 p110$^{-/-}$ Adar2$^{-/-}$* mice (**Fig 4A and 4B, and S1 and S2 Tables**). This drastic reduction in the number of total sites was attributable to the disappearance of intronic sites in *Adar1 p110$^{-/-}$ Adar2$^{-/-}$* mice. We also observed that the number of intronic editing sites was moderately reduced in *Adar1 p110$^{-/-}$* mice. However, although a substantial proportion of the remaining sites showed a reduction in editing ratios, upregulated RNA editing was observed at other sites (**Fig 5A and 5C**), which suggests that ADAR1 p110 and ADAR2 either cooperatively or competitively contribute to intronic editing

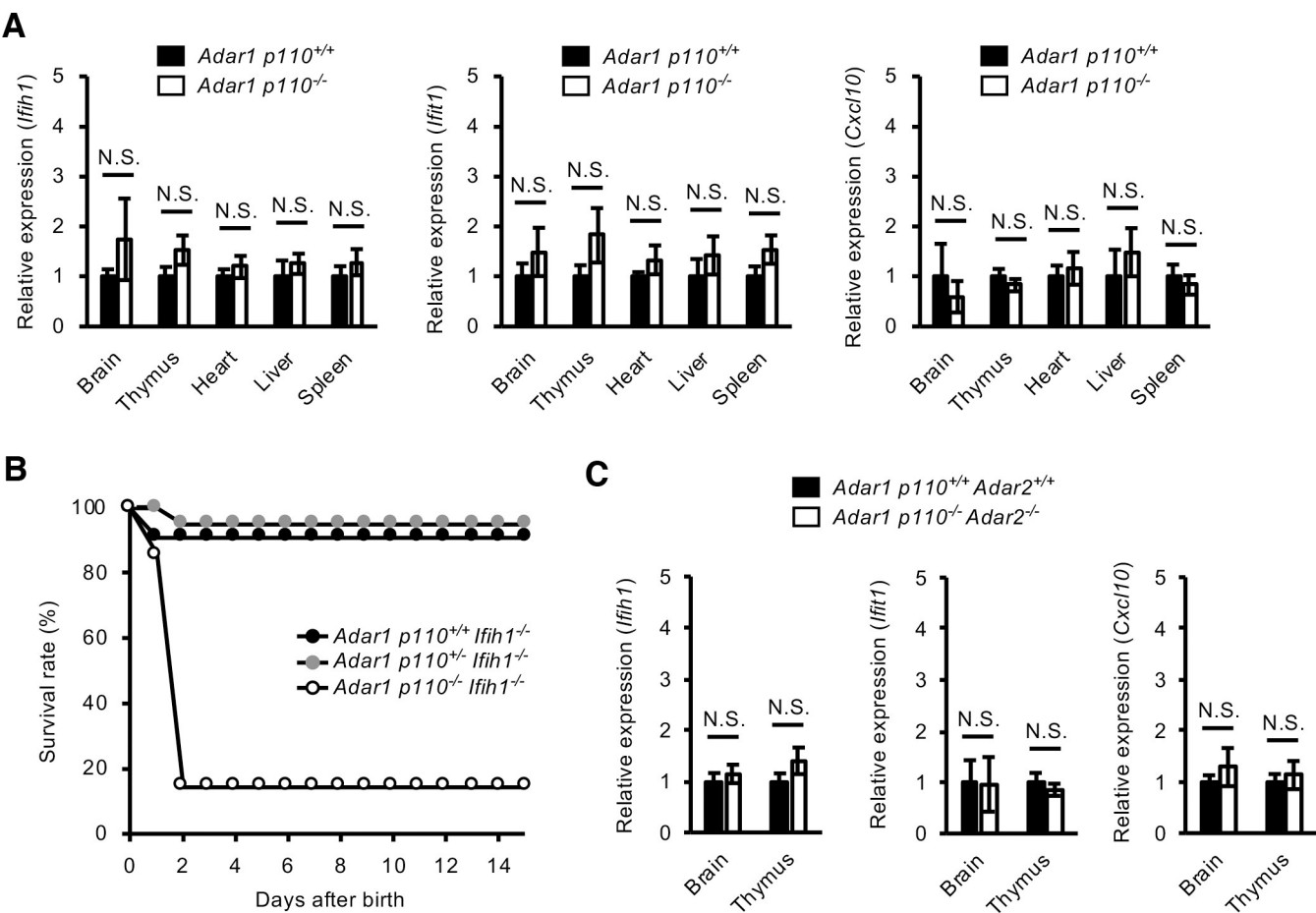

**Fig 3. No activation of MDA5-sensing pathway in *Adar1 p110*–specific KO mice. (A)** The relative expression of the mRNA of the type I interferon-stimulated genes (ISGs), Ifih1 (encoding MDA5), Ifit1, and Cxcl10, in various organs at post-natal day 0 (P0) is compared between wild-type (*Adar1 p110*[+/+]) and *Adar1 p110*–specific knockout (*Adar1 p110*[-/-]) mice. Values represent relative gene expression normalized to GAPDH mRNA and are displayed as the mean ± SEM (n = 4 mice for each group; Mann–Whitney *U*-test, N.S., not significant). **(B)** Survival curves of *Adar1 p110*[+/+] *Ifih1*[-/-] (n = 11), *Adar1 p110*[+/-] *Ifih1*[-/-] (n = 20), and *Adar1 p110*[-/-] *Ifih1*[-/-] mice (n = 7). **(C)** The relative expression of the mRNA of the type I ISGs, Ifih1, Ifit1, and Cxcl10, in the brain and thymus at P0 is compared between wild-type (*Adar1 p110*[+/+] *Adar2*[+/+]), and *Adar1 p110/Adar2* double KO (*Adar1 p110*[-/-] *Adar2*[-/-]) mice. Values represent relative gene expression normalized to GAPDH mRNA and are displayed as the mean ± SEM (n = 4 mice for each group; Mann–Whitney *U*-test, N.S., not significant).

in a site-specific manner. Of note, more than 99% of intronic editing sites found in the brains of *Adar1 p110*[+/+] mice were absent in *Adar1 p110*[-/-] *Adar2*[-/-] mice (**Figs 4A and 5B**). Furthermore, we confirmed that substantially little difference existed in the coverage of the reads at the editing sites among samples examined (**S3 Table**). These data supported the validity of our methodology to identify RNA editing sites, although 17 intronic sites in 15 genes remained in the brains of *Adar1 p110*[-/-] *Adar2*[-/-] mice (**Figs 4A and 5B, and S4 Table**). Therefore, we carefully examined these intronic sites. Among 17 intronic sites, seven sites in five genes were not located in the corresponding intron, which was caused by misregistration of the genes transcribed from the opposite strand in the database. Therefore, we examined RNA editing ratios at the remaining 10 sites in 10 genes, and found no substantial RNA editing in all these sites (**S5 Table**). In contrast, we confirmed RNA editing in some intronic sites except for *Sfi1* in the thymus of *Adar1 p110*[-/-] *Adar2*[-/-] mice (**Fig 5D and S5 Table**). We currently cannot conclude that these intronic sites are edited by ADAR1 p150 in the nucleus. However, the regions containing these intronic sites could be amplified by using not only random hexamers, but also

**A**

### Brain

|  | *Adar1 p110*[+/+] | *Adar1 p110*[-/-] | *Adar1 p110*[-/-] *Adar2*[-/-] |
|---|---|---|---|
| 3'UTR | 382 (191) | 131 (39) | 36 (8) |
| 5'UTR | 6 (3) | 1 (0) | 0 (0) |
| intronic | 6511 (1315) | 2069 (219) | 17 (4) |
| coding | 38 (33) | 27 (21) | 0 (0) |
| others | 1624 (400) | 570 (92) | 63 (14) |
| Total | 8561 (1942) | 2798 (371) | 116 (26) |

**B**

### Thymus

|  | *Adar1 p110*[+/+] | *Adar1 p110*[-/-] | *Adar1 p110*[-/-] *Adar2*[-/-] |
|---|---|---|---|
| 3'UTR | 509 (157) | 475 (155) | 339 (117) |
| 5'UTR | 0 (0) | 0 (0) | 0 (0) |
| intronic | 1219 (129) | 380 (19) | 114 (8) |
| coding | 14 (5) | 6 (2) | 3 (0) |
| others | 372 (68) | 300 (45) | 197 (45) |
| Total | 2114 (359) | 1161 (221) | 653 (170) |

**Fig 4. Number of RNA editing sites in *Adar1 p110*–specific KO and *Adar1 p110/Adar2* double KO mice. (A, B)** The total number and the position of RNA editing sites in brains **(A)** and thymi **(B)** from wild-type (*Adar1 p110*[+/+]), *Adar1 p110*–specific KO (*Adar1 p110*[-/-]), and *Adar1 p110/Adar2* double KO (*Adar1 p110*[-/-] *Adar2*[-/-]) mice at post-natal day 0 (P0) (n = 2 mice for each group) is shown. Values in parentheses are the number of the sites in common between the two mice studied. UTR, untranslated region.

oligo(dT) primers for RT. Also, the editing ratios obtained by oligo(dT) primers were generally equivalent to or slightly higher than those obtained by random hexamers (**S5 Table**). This suggests that RNA editing in these sites might occur in the retained intron mediated by cytoplasmic ADAR1 p150, given that intron retention is frequently observed in mammals [40]. For instance, Fnbp1 mRNA with the retained intron containing the corresponding editing site is registered in the database (**Fig 5E**). Another possibility is that the regions containing such editing sites were amplified from the minor 3'UTR as found in Dlg4 mRNA (**Fig 5F**). Collectively, although we cannot rule out that ADAR1 p150 edits certain intronic sites in the nucleus, these results suggest that the intronic sites are substantially edited by ADAR1 p110 and ADAR2, especially in the mouse brain.

Given that an editing (or exon) complementary sequence (ECS), which is required for RNA editing in coding regions, is usually located in an adjacent intron, these sites are therefore generally edited in the nucleus [21,41]. As expected, editing in the coding sites was not detected in the brains of *Adar1 p110*[-/-] *Adar2*[-/-] mice (**Fig 4A and S2 Table**). Indeed, some coding sites reported to be predominantly edited by ADAR1, including Y/C, Q/R and K/R sites of BLCAP, and A and B sites of serotonin 5-HT$_{2C}$R [21,42–44], showed significant reduction in editing ratios in the brains of *Adar1 p110*[-/-] mice and no editing in those of *Adar1 p110*[-/-] *Adar2*[-/-] mice (**Fig 6A and 6B**). These results indicate that these coding sites are predominantly edited by ADAR1 p110 in the nucleus. In contrast, total RNA-sequencing (RNA-seq) analysis demonstrated that three coding sites in two genes might be edited in the thymus of *Adar1 p110*[-/-] *Adar2*[-/-] mice (**Fig 4B**). Among these sites, we recently reported that two sites of AZIN1 are likely edited by ADAR1 p150 in the cytoplasm using an ECS located in the same exon [21]. Accordingly, we validated RNA editing at two sites of AZIN1 in the thymus of *Adar1 p110*[-/-] *Adar2*[-/-] mice, which was slightly higher than that in wild-type mice but not affected by deletion of either ADAR1 p110 or ADAR2, or both (**Fig 6A and S5 Table**). These results suggest that coding sites that require an intronic ECS are edited in the nucleus, whereas two editing sites in AZIN1 are editable in the cytoplasm because the ECS is likely included in the same exon.

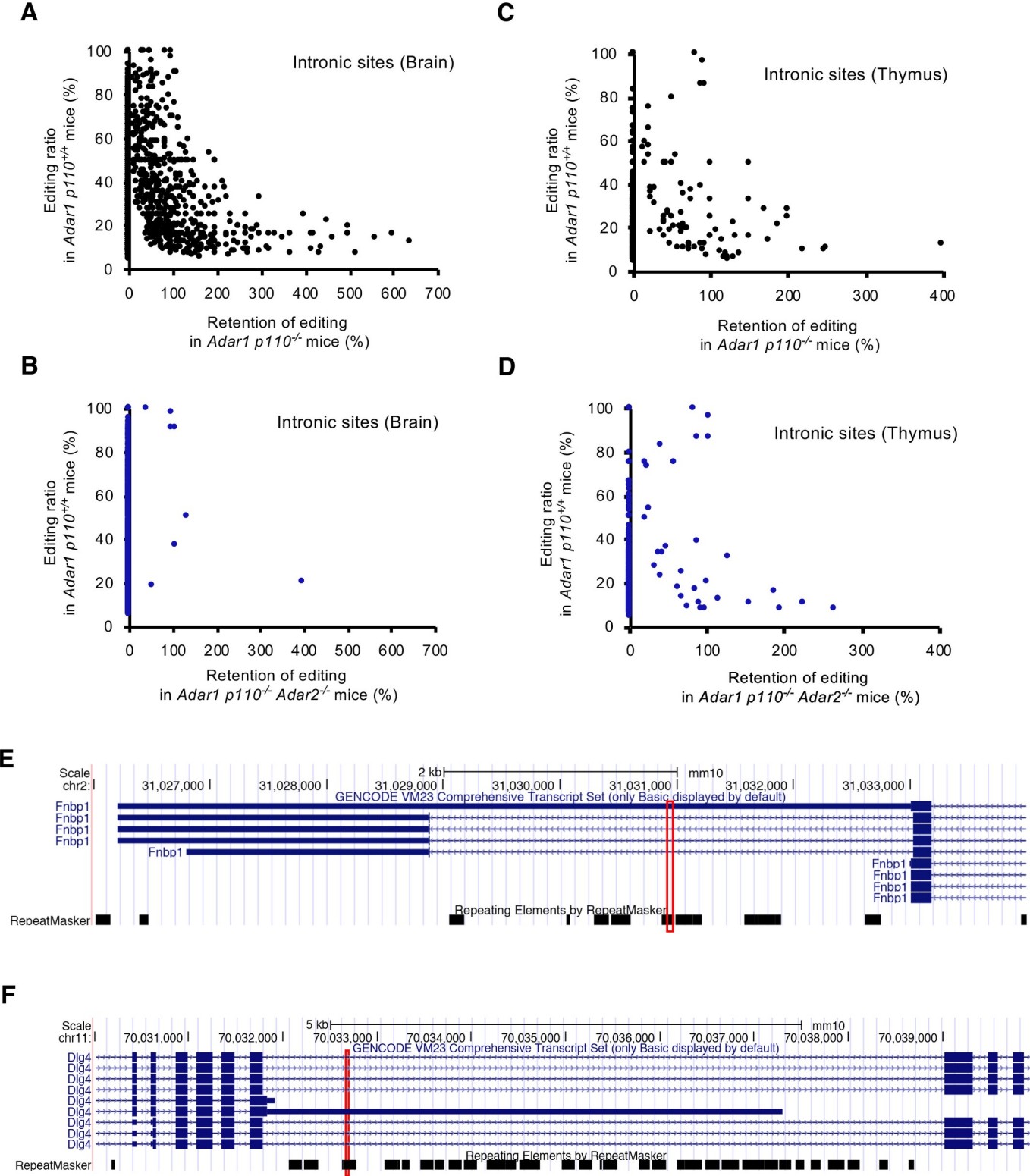

**Fig 5. Retention of RNA editing at intronic sites in *Adar1 p110*–specific KO and *Adar1 p110/Adar2* double KO mice. (A, B)** The mean values for the retention of RNA editing at intronic sites in brains from two *Adar1 p110*-specific KO (*Adar1 p110⁻/⁻*) **(A)** and two *Adar1 p110/Adar2* double KO (*Adar1 p110⁻/⁻ Adar2⁻/⁻*) **(B)** mice at post-natal day 0 (P0). Retention was calculated only for sites where the mean editing ratios of two wild-type (*Adar1 p110⁺/⁺*) mice were more than 5%, which are displayed on the vertical axis. **(C, D)** The mean values for the retention of RNA editing at intronic sites in thymi from two *Adar1 p110⁻/⁻* **(C)** and two *Adar1 p110⁻/⁻ Adar2⁻/⁻* **(D)** mice at P0. Retention was calculated only for the sites where the mean editing ratios of two *Adar1 p110⁺/⁺* mice were more

than 5%, which are displayed on the vertical axis. **(E, F)** The intronic region encompassing RNA editing sites (indicated by a red box) of *Fnbp1* **(E)** and *Dlg4* **(F)** genes was analyzed using the University of California Santa Cruz (UCSC) genome browser.

## ADAR1 p150 contributes to RNA editing at a limited number of sites in the mouse brain

We detected 36 possible RNA editing sites within the 3'UTR in the brains of *Adar1 p110*$^{-/-}$ *Adar2*$^{-/-}$ mice, which accounted for ~9% of the sites found in *Adar1 p110*$^{+/+}$ mice (**Fig 4A and**

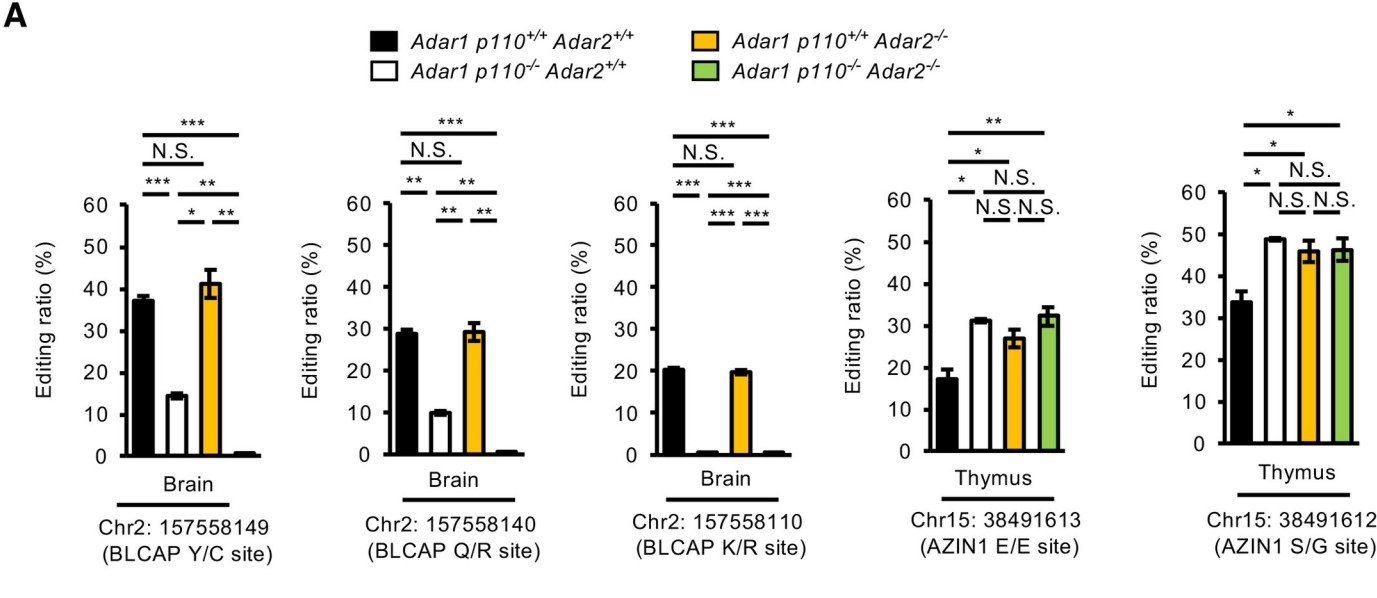

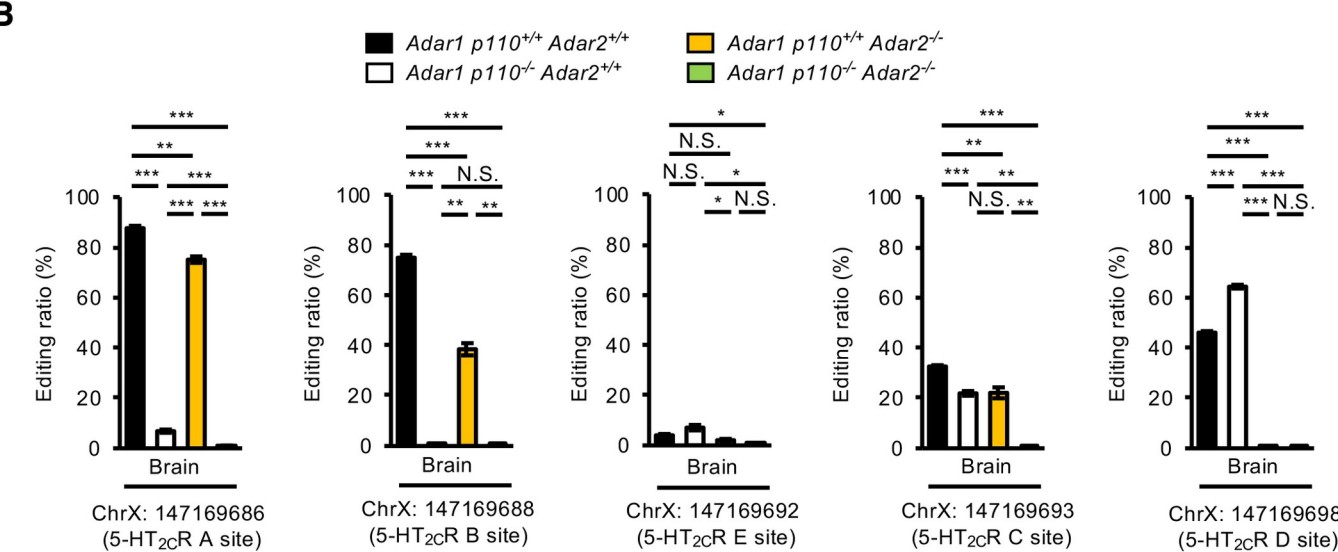

**Fig 6. Comparison of RNA editing ratios at coding sites among *Adar* mutant mice. (A, B)** The editing ratio at each coding site in *Blcap* and *Azin1* genes (**A**) and at the indicated sites of serotonin *Htr2c* gene (**B**), which encodes 5-HT$_{2C}$R, was quantified by Ion S5 sequencing using RNA isolated from indicated organs of wild-type (*Adar1 p110*$^{+/+}$ *Adar2*$^{+/+}$), *Adar1 p110*–specific KO (*Adar1 p110*$^{-/-}$ *Adar2*$^{+/+}$), *Adar2* KO (*Adar1 p110*$^{+/+}$ *Adar2*$^{-/-}$), and *Adar1 p110/Adar2* double KO (*Adar1 p110*$^{-/-}$ *Adar2*$^{-/-}$) mice at post-natal day 0 (P0). Values are displayed as the mean ± SEM (n = 3 mice for each group; Student's *t*-test, *$p < 0.05$, **$p < 0.01$, ***$p < 0.001$, N.S., not significant).

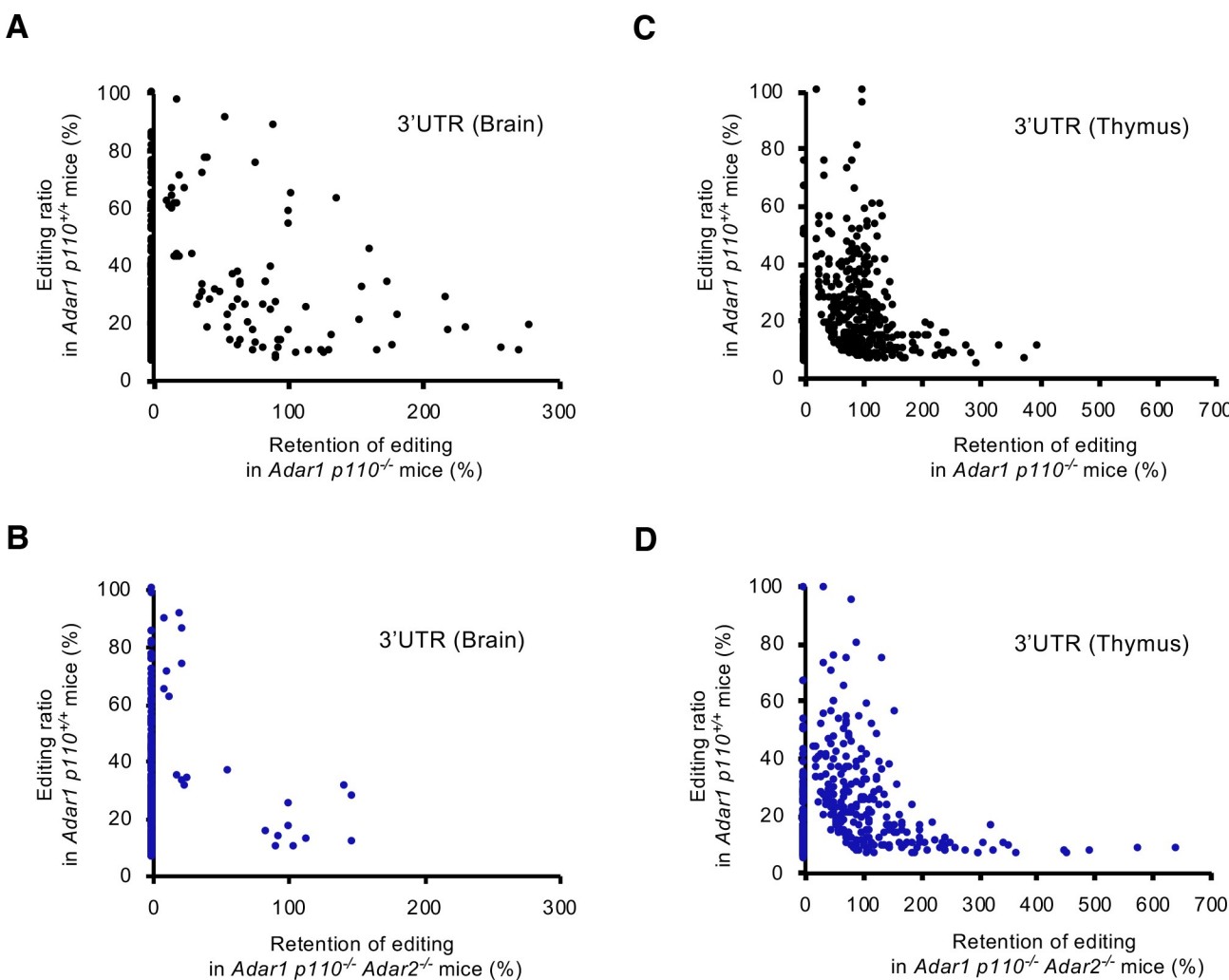

**Fig 7. Retention of RNA editing at sites in 3'UTR in *Adar1 p110*–specific KO and *Adar1 p110/Adar2* double KO mice. (A, B)** The mean values for the retention of RNA editing at sites in the 3'UTR of brains from two *Adar1 p110*–specific knockout (KO; *Adar1 p110⁻/⁻*) **(A)** and *Adar1 p110/Adar2* double KO (*Adar1 p110⁻/⁻ Adar2⁻/⁻*) **(B)** mice at post-natal day 0 (P0). Retention was calculated only for sites where the mean editing ratios of two wild-type (*Adar1 p110⁺/⁺*) mice were more than 5%, which are displayed on the vertical axis. **(C, D)** The mean values for the retention of RNA editing at sites in the 3'UTR of thymi from two *Adar1 p110⁻/⁻* **(C)** and two *Adar1 p110⁻/⁻ Adar2⁻/⁻* **(D)** mice at P0. Retention was calculated only for sites where the mean editing ratios of two *Adar1 p110⁺/⁺* mice were more than 5%, which are displayed on the vertical axis. UTR, untranslated region.

S4 Table). The editing ratios of some of these sites seemed to be sustained under the deficiency of ADAR1 p110 and ADAR2 (Fig 7A and 7B). In contrast, more than two thirds of RNA editing sites in the 3'UTR were detected with a reasonable ratio in the thymus of *Adar1 p110⁻/⁻ Adar2⁻/⁻* mice, which most likely reflects high expression of ADAR1 p150 (Figs 4B, 7C and 7D). Therefore, we validated whether ADAR p150 edited certain sites in the 3'UTR in the mouse brain. Of 36 sites, we first selected four sites in four genes (*Trove2, Cds2, Rpa1,* and *C330018D20Rik*) that were expressed both in the brain and thymus, and quantified the editing ratios as previously described [21]. This analysis demonstrated that all four sites were edited by ~20% in the brains of *Adar1 p110⁻/⁻ Adar2⁻/⁻* mice (Fig 8A and S5 Table), which indicates that ADAR1 p150 contributed to RNA editing in the mouse brain. Intriguingly, the editing ratios of all four sites were not reduced in the absence of either ADAR p110 or ADAR2 or both in the thymus, suggesting that ADAR1 p150 was likely a main contributor to RNA editing of

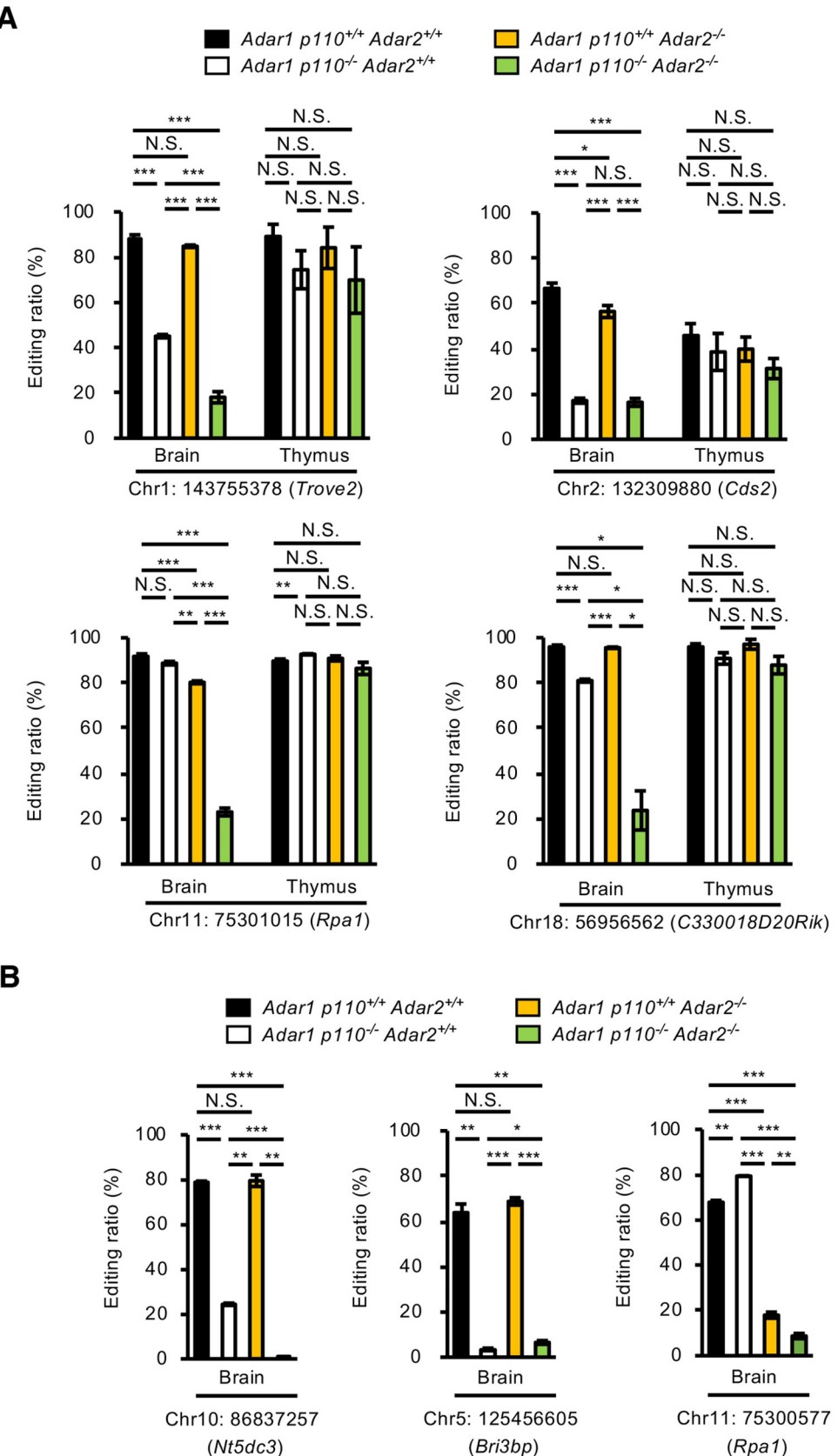

**Fig 8. Comparison of RNA editing ratios at sites in the 3'UTR among various *Adar* mutant mice. (A)** The editing ratios at the indicated sites in the 3'UTR of each gene were quantified by Ion S5 sequencing using RNA isolated from brains and thymi of wild-type (*Adar1 p110*$^{+/+}$ *Adar2*$^{+/+}$), *Adar1 p110*–specific KO (*Adar1 p110*$^{-/-}$ *Adar2*$^{+/+}$), *Adar2* KO (*Adar1 p110*$^{+/+}$ *Adar2*$^{-/-}$), and *Adar1/Adar2* double KO (*Adar1 p110*$^{-/-}$ *Adar2*$^{-/-}$) mice at post-natal day 0 (P0). Values are displayed as the mean ± SEM (n = 3 mice for each group; Student's *t*-test, $^*p < 0.05$, $^{**}p < 0.01$, $^{***}p < 0.001$, N.S., not significant). **(B)** The editing ratios at the indicated sites in the 3'UTR of each gene in the brain at P0 are compared among *Adar1 p110*$^{+/+}$ *Adar2*$^{+/+}$, *Adar1 p110*$^{-/-}$ *Adar2*$^{+/+}$, *Adar1 p110*$^{+/+}$ *Adar2*$^{-/-}$, and *Adar1 p110*$^{-/-}$ *Adar2*$^{-/-}$ mice. Values are displayed as the mean ± SEM (n = 3 mice for each group; Student's *t*-test, $^*p < 0.05$, $^{**}p < 0.01$, $^{***}p < 0.001$, N.S., not significant). UTR, untranslated region.

these sites in the thymus. In contrast, ADAR1 p110 mainly edits sites in *Trove2* and *Cds2*, whereas both ADAR1 p110 and ADAR2 can edit sites in *Rpa1* and *C330018D20Rik* in a mutually complementary manner (**Fig 8A**). These results suggest that RNA editing in the 3'UTR was catalyzed by both ADAR1 p110 and ADAR2 in the nucleus in a site-specific manner, and subsequently further catalyzed by cytoplasmic ADAR1 p150. We examined three more sites in three genes (*Nt5dc3*, *Bri3bp*, and *Rpa1*) selected from the list of sites where RNA editing was not detected in the brains of *Adar1 p110*$^{-/-}$ *Adar2*$^{-/-}$ mice (**S2 Table**) to better understand the contribution of each ADAR. Sites in *Nt5dc3* and *Bri3bp* were mainly edited by ADAR1 p110, which was similar to those in *Trove2* and *Cds2* (**Fig 8A and 8B**). In contrast, the Chr11:75300577 site in *Rpa1* was predominantly edited by ADAR2, which differs from the nearby Chr11:75301015 site, which both ADAR1 p110 and ADAR2 could highly edit (**Fig 8A and 8B**). These results collectively suggest that ADAR1 p110, ADAR1 p150 and ADAR2 contribute to RNA editing in the 3'UTR in a site- and organ-specific manner, depending on the expression level of each ADAR, at least in part.

## Some selective sites in the 3'UTR are efficiently edited in the brains of *Adar1 p110*$^{-/-}$ *Adar2*$^{-/-}$ mice

We further selected four sites in the 3'UTR of three genes (*Fubp3*, *Trim12c*, and *Car5b*) among 36 sites left in the brain of *Adar1 p110*$^{-/-}$ *Adar2*$^{-/-}$ mice. Intriguingly, the editing ratios of these sites were preserved or slightly upregulated in a deficiency of either ADAR1 p110 or ADAR2, or both (**Fig 9A**). Furthermore, we found that the editing ratios of these sites were preserved in the brain of *Adar1*$^{E861A/p110del}$ mice, in which active ADAR1 p150 is expressed from one allele, which suggests that a subtle amount of ADAR1 p150 is sufficient to sustain the editing ratio of certain sites, which are thereby likely ADAR1 p150–specific sites. However, all these sites were edited by less than 50%, which might not be enough to escape MDA5-sensing. Therefore, we further examined highly edited sites in the presence of only ADAR1 p150 and found sites edited by more than ~60% in *Mad2l1* gene in the brains of *Adar1 p110*$^{-/-}$ *Adar2*$^{-/-}$ mice (**Fig 9B and S5 Table**). The Chr6:66540081 site of *Mad2l1* was edited ~70% and located in the long dsRNA structure along with many other sites; most of these were edited by less than 10% (**Fig 9A and 9B**). Taken together, the results indicated that certain sites in the 3'UTR are efficiently edited by a very limited amount of ADAR1 p150 in the mouse brain, which might affect the dsRNA structure required to prevent MDA5 activation.

## Discussion

In this study, we generated *Adar1 p110*–specific KO mice, which displayed a high mortality rate during early post-natal developmental stages. It has been demonstrated that *Adar1* KO and *Adar1 p150*–specific KO mice show embryonic lethality, which can be rescued by concurrent deletion of either MDA5 or MAVS [33,34]. However, although more than half of *Adar1 p150/Mavs* double KO mice can survive after weaning, *Adar1/Mavs* double KO and *Adar1/*

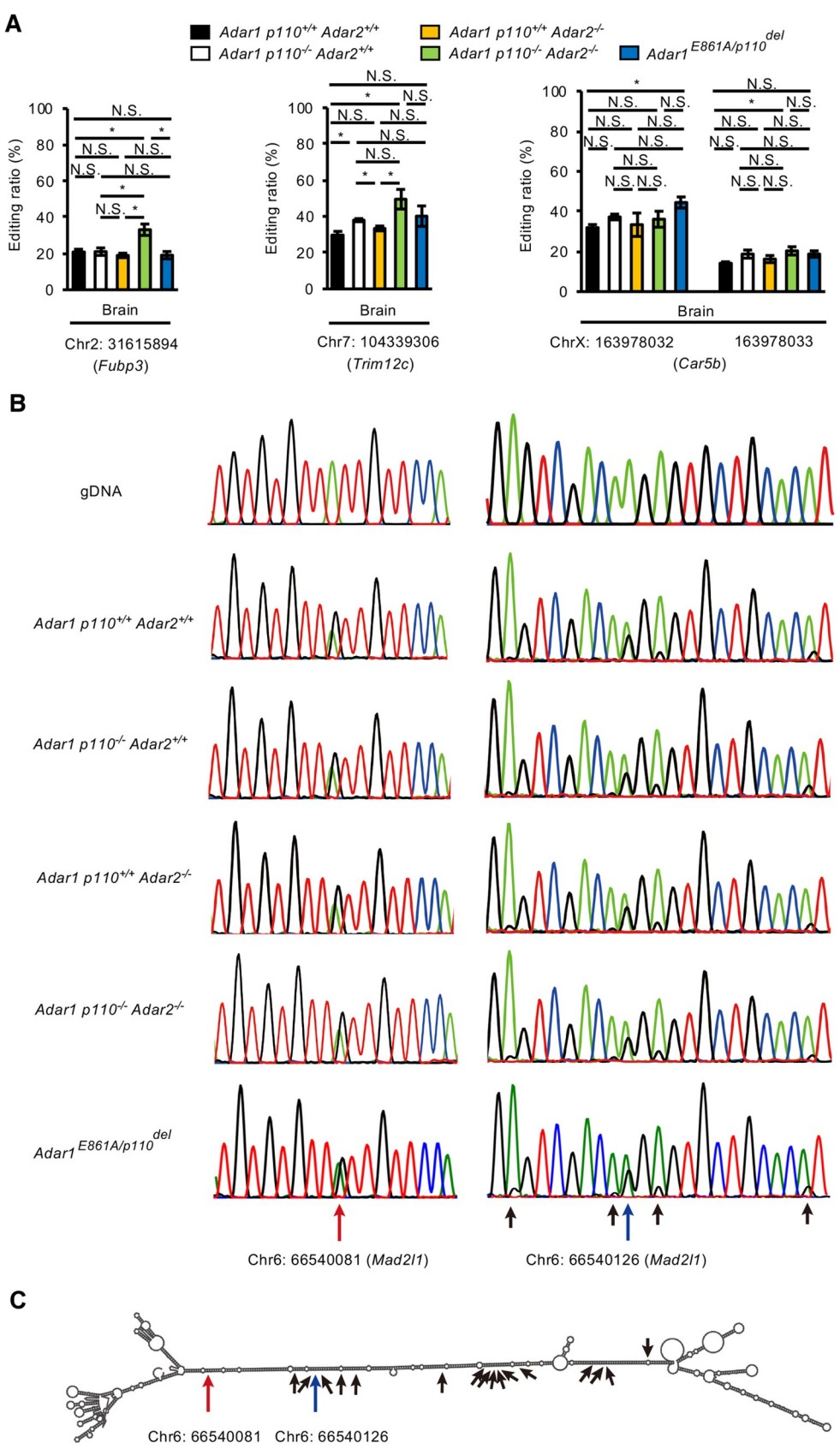

**Fig 9. RNA editing sites preserved in *Adar1 p110/Adar2* double KO mice. (A)** Editing ratios at indicated sites in the 3'UTR of each gene in the brain at post-natal day 0 (P0) are compared among wild-type (*Adar1 p110^+/+ Adar2^+/+*), *Adar1 p110*–specific knockout (KO; *Adar1 p110^-/- Adar2^+/+*), *Adar2* KO (*Adar1 p110^+/+ Adar2^-/-*), *Adar1 p110/Adar2* double KO (*Adar1 p110^-/- Adar2^-/-*), and *Adar1^E861A/p110del* mice. Values are displayed as the mean ± SEM (n = 3 mice for each group; Student's *t*-test, *$p < 0.05$, N.S., not significant). **(B)** Representative chromatograms of the Sanger sequencing of PCR products derived from the *Mad2l1* gene in brains of the indicated mutant mice at P0 are displayed. The highly edited sites (> 60%) in *Adar1 p110/Adar2* double KO mice are indicated with long red arrows, while other edited sites are indicated with blue (30–60%) or black (<30%) arrows. **(C)** The secondary structure of a part of the 3'UTR of the *Mad2l1* gene was estimated using the RNAfold web server. The highly edited sites (> 60%) are indicated with long red arrows, while other edited sites are indicated with blue (30–60%) or black (<30%) arrows. UTR, untranslated region.

*Ifih1* double KO mice die just after birth. We revealed that this difference is attributable to a role for ADAR1 p110 in perinatal stages, at least in part. Furthermore, given that *Adar1^E861A/E861A Ifih1* KO mice can survive until adulthood [20,21,30,36], it is postulated that ADAR1 has RNA editing–independent functions. In this regard, we showed that the high mortality rate found in *Adar1 p110*–specific KO mice could be rescued by the expression of inactive ADAR1 but not the concurrent deletion of MDA5, which indicates that ADAR1 p110 has an additional function other than RNA editing. Although it has been proposed that ADAR1 is involved in microRNA processing, and mRNA surveillance and decay in an RNA editing–independent manner [45–50], further investigation is needed to specify which function is critical *in vivo*. In addition, we currently cannot identify which organs require such an RNA editing–independent function to avoid a high mortality rate. Abnormal pathologies can be found in multiple organs, including the kidney and intestines in surviving *Adar1/Mavs* double KO mice [34]. However, evident abnormalities could not be observed in such organs of *Adar1 p110*–specific KO mice. Considering that ~20% of *Adar1 p110*–specific KO mice can be mature and fertile, the development of certain essential functions, such as suckling and breathing, might be slightly delayed, originated from a certain restricted region in the brain. Therefore, the role of ADAR1 p110 in neurons should be investigated further, for instance, by creating neuron-specific *Adar1* or *Adar1 p110* KO mice in future.

Many questions exist concerning the regulatory mechanism underlying the expression of the two isoforms of ADAR1 *in vivo*. For instance, it remains unknown how two promoters contribute to the expression of ADAR1 p150. In addition, ADAR1 p110 might be translated from Adar1 p150 mRNA using a downstream start codon located in exon 2 [39]. Furthermore, ADAR1 p150 might contribute to RNA editing in the nucleus by shuttling between the nucleus and cytoplasm [17,19]. This study elucidated that the expression of ADAR1 p150 can be maintained without a constitutive promoter, suggesting that the transcription of each isoform is most likely regulated under a different promoter. In addition, ADAR1 p110 is barely detectable in *Adar1 p110*–specific KO mice, even in the thymus where ADAR1 p150 is abundantly expressed. Furthermore, ADAR1 p110 was not apparently induced in the splenocytes of *Adar1 p110*–specific KO mice by the treatment with IFN, which upregulated ADAR1 p150 protein expression. This demonstrates that ADAR1 p110 is not efficiently translated from Adar1 p150 mRNA in mice, which might be different from the phenomena reported in human cell lines [46,51,52]. Furthermore, given that intronic sites were substantially absent in the brain of *Adar1 p110/Adar2* double KO mice, ADAR1 p150 does not contribute to RNA editing in the nucleus, at least in the mouse brain. Therefore, each isoform likely plays a distinct function in a different intracellular compartment.

The overexpression of ISGs is observed in the mouse brain in the absence of ADAR1 activity [36], indicating that ADAR1–mediated RNA editing is required to prevent MDA5 activation in this organ. Furthermore, the loss of ADAR1 p150 activates the MDA5 signaling pathway [34], whereas we showed that a loss of ADAR1 p110 does not induce MDA5 activation. In addition, this study revealed that the embryonic lethality found in *Adar1^E861A/E861A* mice can be rescued by

the sole expression of ADAR1 p150. Of note, we and other groups have recently reported that a point mutation in the ADAR1 p150–specific Zα domain induces the upregulated expression of ISGs in multiple organs, including the brain [53–56]. These lines of evidence indicate that ADAR1 p150–mediated RNA editing is essential to escape MDA5-sensing, even in brain where a subtle amount of ADAR1 p150 is expressed. However, only 36 RNA editing sites were identified within the 3'UTR in the brains of *Adar1 p110*/*Adar2* double KO mice. In addition, most of these sites were edited by less than 40%, which is questionable with regard to exerting a significant effect on escaping MDA5-sensing. We have currently identified one highly edited region in the 3'UTR of *Mad2l1*, which might affect the stability of a dsRNA structure. Although this region is a prime candidate in terms of being edited in the mouse brain to prevent MDA5 activation, additional highly edited sites might exist, for instance, in the category of "others" involving 63 sites, which likely originated from unidentified transcripts or rare longer 3'UTRs. Furthermore, due to technical limitations, certain hyperedited sites are sometimes difficult to identify [30,57]. It is noteworthy that the editing ratios of certain sites in the 3'UTR are not altered in the absence of either ADAR1 p110 or ADAR2, or both, although they are highly expressed in the brain. This evidence suggests the presence of ADAR1 p150–specific sites. Given that dsRNA-binding domains and a deaminase domain are common between ADAR1 p110 and p150 [6–8], the difference in target recognition between these two isoforms has not received much attention. Nevertheless, the ADAR1 p150–specific Zα domain can bind to both Z-DNA and Z-RNA [58–60]. In addition, ADAR1 p150 can edit dsRNA more efficiently with Z-RNA *in vitro* [61]. Furthermore, we have recently shown that this unique domain contributes to efficient editing of certain sites [53]. Therefore, the continual identification of highly edited sites in the brains of *Adar1 p110*/*Adar2* double KO mice is required to identify the ADAR1 p150–specific sites critical for preventing MDA5 activation.

We currently do not know the reason why ADAR1 p150 is expressed at the lowest level in the mouse brain. Considering that the expression level of MDA5, which is regulated by IFN as an ISG, is also relatively low in the brain, a positive correlation might exist between the expression level of these two proteins. In this case, the same dsRNAs might be recognized as non-self in certain organs but not in others, depending on the level of RNA editing and the amount of MDA5. Of note, ADAR1–mediated RNA editing is also essential for maintaining homeostasis in the human brain. Loss-of- and gain-of-function mutations have been identified in *ADAR1* and *IFIH1* genes, respectively, in patients with AGS, in which the brain is one of the most affected organs accompanied by a type I IFN signature [37]. *ADAR1* mutations in patients with AGS are frequently located in the catalytic domain of the protein, leading to reduced RNA-editing activity [38]. In addition, another mutation is found in the p150 isoform–specific Zα domain, which indicates that the reduced RNA editing activity of ADAR1 p150 is probably a cause of AGS pathogenesis [62,63]. Indeed, we have recently found that a point mutation in the Zα domain induces AGS-like encephalopathy in mice [53]. Therefore, the minimal expression of ADAR1 p150 under healthy conditions might be a cause of severer damage in the brain in response to the overproduction of IFN as found in patients with AGS. Understanding the mechanisms underlying ADAR p150–mediated RNA editing will lead to the elucidation of disease pathogenesis and the establishment of a treatment for AGS.

## Materials and methods

### Ethics statement

All experimental procedures that included mice were performed following with protocols approved by the Institutional Animal Care and Use Committee of Osaka University (27-004-023 and 01-063-005).

## Mice maintenance

Mice were maintained on a 12-h light/dark cycle at a temperature of 23 ± 1.5˚C with a humidity of 45 ± 15% as previously described [50].

## Mutant mice

*Ifih1*[-/-], *Adar2*[-/-] *Gria2*[R/R], and *Adar1*[E861A/E861A] mice were maintained in our laboratory as previously described [21]. *Adar1*[M249A/M249A] mice and *Adar1 p110*[-/-] mice were generated by genome editing using a CRISPR/Cas9 system at the Genome Editing Research and Development Center, Graduate School of Medicine, Osaka University. Briefly, to create *Adar1*[M249A/M249A] mice, guide RNA targeting 5'-ACCTTCTGAGCCTCTTGACA-3' was synthesized by using GeneArt Precision gRNA Synthesis Kit (Thermo Fisher Scientific). Pronuclear-stage mouse embryos (CLEA Japan Inc., Tokyo, Japan) were electroporated to introduce Cas9 mRNA, the guide RNA and a single-stranded donor (5'- TGAGGACGGAGACCCTGCCT CTGACTTAGAAGGACCTTCTGAGCCTCTTGAC<u>GCC</u>GCTGAAATCAAGGAGAAG ATCTGTGACTATCTGTTCAATGT-3'); these introduced a point mutation at the corresponding codon (underlined GCC in the target nucleotide). To create *Adar1 p110*[-/-] mice, two Alt-R CRISPR-Cas9 CRISPR RNA (crRNAs; CrRNA-Adar1p110-KO-up [5'-ATAGTGATCG TAATCACACT-3'] and CrRNA-Adar1p110-KO-down [5'-GGCAAGAGTGCAATCAGC CG-3']) were synthesized at Integrated DNA Technologies (IDT) and hybridized with trans-activating CRISPR RNA (tracrRNA), generating guide RNAs, which were introduced into pronuclear-stage mouse embryos with Cas9 mRNA by electroporation. Mouse embryos that developed to the two-cell stage were transferred into the oviducts of female surrogates. *Adar1 p110*[-/-] *Adar2*[-/-] mice were obtained by crossing *Adar1 p110*[+/-] *Adar2*[+/-] mice and sacrificed at P0 to obtain total RNAs. We further created *Adar1 p110*[-/-] *Adar2*[-/-] *Gria2*[R/R] mice by repeated backcrossing to induce homologous recombination, given that the *Adar1* and *Gria2* genes localize to the same chromosome.

For genotyping of *Adar1*[M249A/M249A] mice, genomic DNA was amplified with the following primers: 5'-GAAGGGAAAGCTGCACAGAGGAA-3' and 5'-TACTTTCATGCTTTATCGC AGGCTC-3'. After treating with illustra ExoProStar (GE Healthcare), PCR products were directly sequenced using the primer 5'-CTCCTTTGTGGAGCCTTGTG-3'. Genotyping PCR of samples from *Adar1 p110*[-/-] mice was performed using the following three primers: 5'-GAC CCATCAGTCTTGCATCTAGATA-3', 5'-GCATTTACACAGCTACTACATTGCT-3' and 5'-TAATGACCTGCTCTAACAACCTGTC-3', which generated a 333-bp fragment for a wild-type allele and a 180-bp fragment for a mutated allele. All mice used in experiments were in a C57BL/6J background.

## Cell culture

A single-cell suspension of splenocytes was obtained by mashing an isolated spleen through a 70-μm cell strainer (Falcon) followed by lysis of erythrocytes with RBC lysis buffer (BioLegend) as previously described [50]. Then, splenocytes were cultured in Dulbecco's modified Eagle's medium (Nacalai Tesque) containing 10% feral bovine serum and 1% penicillin/streptomycin (Nacalai Tesque) in the presence or the absence of IFN-β1 (100 ng/ml, BioLegend) for 20 h. Cells were maintained at 37˚C in the presence of 5% $CO_2$.

## Preparation of tissue lysates

Tissue lysates were prepared as described previously with some modifications [64]. In brief, isolated organs were frozen in liquid nitrogen, thawed once at room temperature, and then

homogenized in lysis buffer (0.175 M Tris-HCl, pH 6.8, 3% SDS, and 5 mM EDTA). After boiling at 95°C for 10 min, the lysates were passed through a 23-gauge needle followed by centrifugation at 20,000 × g at 4°C for 10 min. Each supernatant was transferred to a 1.5 mL tube and stored at -80°C until use.

## Subcellular fractionation

To perform cytoplasmic and nuclear fractionation, isolated splenocytes were incubated in a hypotonic buffer (20 mM HEPES, pH 7.4, 10 mM KCl, 2 mM $MgCl_2$, 1 mM EDTA, 1 mM EGTA, 1 mM DTT, and 1× Protease Inhibitor) on ice for 15 min. The nuclei were separated from soluble cytoplasmic fraction by centrifugation at 800 × g at 4°C for 5 min. The supernatant was then centrifuged at 10,000 × g at 4°C for 10 min. The resulting supernatant was transferred to a new tube as the cytoplasmic fraction. The nuclear pellet was washed twice with the hypotonic buffer containing 0.1% NP-40 and suspended with RIPA buffer (50 mM Tris-HCl, pH 7.4, 150 mM NaCl, 1% Triton X-100, 1 mM EDTA, 0.1% SDS, 0.5% deoxycholate). After sonication (pulse 20%, 10sec, 3 times) using a ultrasonic homogenizer (MitsuiElectric), the mixture was centrifugated at 10,000 × g at 4°C for 10 min. The supernatant was transferred to a new tube as the nuclear fraction. Total cell lysates were prepared by suspending splenocytes with RIPA buffer followed by sonication and centrifugation as described above.

## Immunoblot analysis

All the samples were dissolved in SDS-PAGE sample buffer and boiled at 95°C for 5 min prior to immunoblotting assay. Immunoblot analysis was performed as previously described [21,23] with minor modifications. In brief, each protein lysate sample was separated by sodium dodecyl sulfate–polyacrylamide gel electrophoresis, transferred to a polyvinylidene difluoride membrane (Bio-Rad), and immunoblotted at 4°C overnight with primary antibodies: mouse monoclonal anti-ADAR1 antibody (15.8.6; Santa Cruz Biotechnology), mouse monoclonal anti-ADAR2 antibody (1.3.1; Santa Cruz Biotechnology), rabbit monoclonal anti-MDA5 antibody (D74E4; Cell Signaling Technology), rabbit polyclonal anti-Lamin B1 antibody (12987-1-AP; Proteintech), and mouse monoclonal anti-GAPDH antibody (3H12; MBL).

## Total RNA extraction

Total RNA was extracted from isolated organs using TRIzol reagent (Thermo Fisher Scientific) by following the manufacturer's instructions. After measuring the RNA concentration using a NanoDrop One (Thermo Fisher Scientific), total RNA samples were stored at -80°C until use.

## Total RNA sequencing analysis

Total RNA sequencing (RNA-seq) analyses were performed at Macrogen (Kyoto, Japan), including removal of ribosomal RNAs and library preparation. The library samples were subjected to deep sequencing using an Illumina NovaSeq 6000 with 100-bp paired-end reads.

## Genome-wide identification of RNA editing sites

We adopted a genome-wide approach to identify editing sites with total RNA-seq reads as previously described [21,23] but with modifications, based in part on the literature [30,65]. In brief, sequence reads were mapped onto a reference mouse genome (GRCm38/mm10) with a spliced aligner HISAT2 [66]. The mapped reads were then processed by adding read groups, and sorting and marking duplicates with the tools AddOrReplaceReadGroups and MarkDuplicates compiled in GATK4 [67]. GATK SplitNCigarReads, BaseRecalibrator, and

ApplyBQSR were used to split 'N' trim and reassign mapping qualities, which output analysis-ready reads for subsequent variant calling. The GATK HaplotypeCaller was run for variant detection, in which the stand-call-conf option was set to 20.0 and the dont-use-soft-clipped-bases option was used. The results of variant calling were further filtered by GATK VariantFiltration using Fisher strand values (FS) > 30.0 and quality by depth values (QD) < 2.0 as recommended by the GATK developer for RNA-seq analysis. The remaining variants that were expected to be of high quality were annotated with ANNOVAR software [68], where gene-based annotation was generated with a RefSeq database [69]. Among these variants, we picked up known editing sites registered in the following databases: database of RNA editing (DARNED) [70], rigorously annotated database of A-to-I RNA editing (RADAR) [71], or REDIportal [72]. Of note, DARNED and REDIportal provide editing sites in mm10 coordinates, while RADAR does in mm9. This discordance was resolved in a way that RADAR-registered editing sites were uplifted to mm10 coordinates with a liftOver tool in University of California Santa Cruz (UCSC) Genome Browser utilities [73]. It should also be noted that the strand orientation in the predicted variants was strictly checked with that of the database annotations. Finally, A-to-I editing ratios in each sample were calculated by dividing the allelic depth by the read depth for the editing sites shown in the annotated results as recently described [53].

### Calculation of editing retention

To determine how much editing was retained in tissues from *Adar1 p110*$^{-/-}$ and *Adar1 p110*$^{-/-}$ *Adar2*$^{-/-}$ mice, the mean editing ratio of each mutant mouse (n = 2) was calculated by using the ratios obtained by total RNA-seq analysis (S2 Table). Then, each ratio was divided by the mean editing ratio of *Adar1 p110*$^{+/+}$ mice (n = 2) to calculate the value for the retention of editing at each site as previously described [21]. We considered only sites with a more than 5% mean editing ratio in *Adar1 p110*$^{+/+}$ mice and ones detected in both two mice of each group.

### Quantification of RNA editing ratio with Ion amplicon sequencing reads

The preparation of Ion amplicon libraries for quantification of RNA editing sites has been previously described [21]. In brief, 1 μg of total RNA was treated with 0.1U/μL DNase I (Thermo Fisher Scientific) at 37˚C for 15 min and denatured at 65˚C for 15 min. Complementary DNA (cDNA) was synthesized by RT using a Superscript III First-Strand Synthesis System (Thermo Fisher Scientific) with oligo(dT) primers or random hexamers. A first round of PCR was performed using cDNA, Phusion Hot Start High-Fidelity DNA Polymerase (Thermo Fisher Scientific), and first primers that were editing-site specific (S5 Table). A second round of PCR was then performed using an aliquot of the first PCR product as a template and second primers that were editing-site specific; an A adaptor (5'-CCATCTCATCCCTGCGTGTCTCCGACT CAG-3'), an Ion Xpress Barcode and a trP1 adaptor (5'-CCTCTCTATGGGCAGTCGGTGA T-3') were in forward and reverse primers, respectively (S5 Table). After gel purification, the concentration of each PCR product was measured using a NanoDrop One and then equal amounts of 40–170 PCR products were combined. After a quality check using a 2100 Bioanalyzer (Agilent Technologies) with a High Sensitivity DNA kit, the resultant amplicon library samples were subjected to deep sequencing using an Ion S5 system (Thermo Fisher Scientific) at the CoMIT Omics Center, Graduate School of Medicine, Osaka University. RNA editing ratios were calculated using an in-house program as previously described [21].

### qRT–PCR analysis

As previously described [23,50], cDNA was synthesized from total RNA extracted from various organs using a ReverTra Ace qPCR RT Master Mix with gDNA Remover (Toyobo). The

qRT–PCR reaction mixture was prepared by combining each target-specific primers and probes with a THUNDERBIRD Probe qPCR Mix (Toyobo). The qRT–PCR was performed using an ABI Prism 7900HT Real-Time PCR System (Applied Biosystems). The sequences of primers and probes for *Adar1 p150*, *Ifih1*, *Ifit1*, *Cxcl10*, and *GAPDH* has been previously reported [23]. The expression level of each mRNA relative to that of GAPDH mRNA was calculated by the ΔΔCt method.

## Direct Sanger sequencing analysis

After RT with oligo(dT) primer, PCR was performed with Phusion Hot Start High-Fidelity DNA Polymerase and the following primers: Mad2l1 site1 Fw (5'-ATGGTGGCCTACAAAA CCCC-3') and Mad2l1 site1 Rv (5'-ACTGGATCCGGATTGGCAAC-3') for *Mad2l1* site1, Mad2l1 site2 Fw (5'-AGGGAAGAATCAGGACCCCA-3') and Mad2l1 site2 Rv (5'-GCATC AACTGCTTTGTGAGC-3') for *Mad2l1* site2. After gel purification, each RT–PCR product was directly sequenced using the following primers: Mad2l1 site1 Seq (5'-TCCCAAGTGTG CTTTTCCAGA-3') for *Mad2l1* site1, Mad2l1 site2 Seq (5'-TGATTTACAACCACAAGCCA CA-3') for *Mad2l1* site2. The editing ratio was estimated as the % ratio of the "G" peak over the sum of the "G" and "A" peaks of the sequencing chromatogram as previously described [21,64].

## Analysis of dsRNA structure

Potential secondary dsRNA structures were calculated using the RNAfold web server [74].

## Histological analysis

Embryos at E15.5 and various organs isolated at P0 were fixed in 10% formalin neutral buffer solution (Wako) with shaking at 4˚C for 2 days. Hematoxylin and eosin (HE) staining, and morphological analysis by pathologists were performed at New Histo. Science Laboratory Co. Ltd. (Tokyo, Japan), Morphotechnology Co. Ltd. (Hokkaido, Japan) and Applied Medical Research Laboratory Co. Ltd. (Osaka, Japan). Nissl staining was also performed for brain samples.

## Statistical analysis

Either a Mann-Whitney *U*-test or Student's *t*-test was used as indicated in each figure legend. All values are displayed as the mean ± standard error of the mean (SEM). Non-significance is displayed as N.S., while statistical significance is displayed as $p < 0.05$ (*), $p < 0.01$ (**) or $p < 0.001$ (***).

## Supporting information

**S1 Fig. Generation of *Adar1*$^{M249A/M249A}$ mice. (A)** Schematic diagram of mouse *Adar1* gene. A point mutation was inserted at the position of amino acid 249 from ATG to GCC in exon 2 (Ex2), which converted the initiation methionine (M249) to alanine (A). This conversion was expected to specifically inhibit translation of the ADAR1 p110 isoform but instead induced the expression of a truncated p110 isoform translated from the downstream methionine most likely located in either Ex2 or Ex3. **(B)** The number of the surviving littermates of wild-type (*Adar1*$^{+/+}$), *Adar1*$^{M249A/+}$, and *Adar1*$^{M249A/M249A}$ mice between 4 to 6 weeks of age. **(C)** Immunoblot analysis of ADAR1 p110 and p150 protein expression in brains, thymi, and spleens of *Adar1*$^{+/+}$, *Adar1*$^{M249A/+}$, and *Adar1*$^{M249A/M249A}$ mice. The expression of GAPDH protein is

shown as a reference.
(TIF)

**S2 Fig. The expression and localization of ADAR1 isoforms in *Adar1 p110*–Specific KO mice. (A)** Immunoblot analysis of ADAR1 p110 and ADAR1 p150 protein expression in total lysates (T), nuclear fraction (N), and cytoplasmic fraction (C) of splenocytes isolated from wild-type (*Adar1 p110*$^{+/+}$) and *Adar1 p110*–specific knockout (KO; *Adar1 p110*$^{-/-}$) mice. The expression of Lamin B1 and GAPDH proteins is shown as nuclear and cytoplasmic markers, respectively. **(B)** Immunoblot analysis of ADAR1 p110 and ADAR1 p150 protein expression in splenocytes isolated from *Adar1 p110*$^{+/+}$ and *Adar1 p110*$^{-/-}$ mice. Splenocytes were cultured in the absence (-) or the presence (+) of interferon (IFN)-β1 stimulation for 20 h. The expression of Lamin B1 protein is shown as reference.
(TIF)

**S3 Fig. Morphological analysis of *Adar1 p110*–Specific knockout mice.** Representative images of hematoxylin and eosin (HE) staining of the whole embryo at E15.5 and the brain, lung, heart, liver, kidney and intestine at post-natal day 0 (P0), and those of Nissl staining of the brain at P0 from wild-type (*Adar1 p110*$^{+/+}$) and *Adar1 p110*–specific knockout (*Adar1 p110*$^{-/-}$) mice. The scale bar is indicated in each panel.
(TIF)

**S1 Table. Total number of reads for each sample obtained by RNA-seq.**
(TIF)

**S2 Table. List of RNA editing sites identified by total RNA-seq analysis.**
(XLSX)

**S3 Table. Coverage of the reads at the editing sites identified in this study.**
(TIF)

**S4 Table. List of RNA editing sites in intron and 3'UTR identified in the brain of *Adar1 p110/Adar2* double KO mice by total RNA-seq analysis.**
(XLSX)

**S5 Table. Quantified RNA editing ratios with information on primer sequences used for Ion amplicon libraries.**
(XLSX)

## Acknowledgments

We thank all staff in the Genome Editing Research and Development Center, the Center for Medical Research and Education, the CoMIT Omics Center, and the Institute of Experimental Animal Sciences, Graduate School of Medicine, Osaka University, for technical support. Computations were partially performed on the NIG supercomputer at ROIS National Institute of Genetics, Japan.

## Author Contributions

**Conceptualization:** Taisuke Nakahama, Yuki Kato, Yukio Kawahara.

**Data curation:** Taisuke Nakahama, Toshiharu Shibuya, Yuki Kato, Yukio Kawahara.

**Formal analysis:** Jung In Kim, Taisuke Nakahama, Ryuichiro Yamasaki, Pedro Henrique Costa Cruz, Tuangtong Vongpipatana, Maal Inoue, Nao Kanou, Yanfang Xing, Hiroyuki Todo, Toshiharu Shibuya, Yuki Kato, Yukio Kawahara.

**Funding acquisition:** Jung In Kim, Taisuke Nakahama, Tuangtong Vongpipatana, Yuki Kato, Yukio Kawahara.

**Investigation:** Jung In Kim, Taisuke Nakahama, Ryuichiro Yamasaki, Tuangtong Vongpipatana, Maal Inoue, Yanfang Xing, Toshiharu Shibuya, Yuki Kato, Yukio Kawahara.

**Methodology:** Taisuke Nakahama, Pedro Henrique Costa Cruz, Toshiharu Shibuya, Yuki Kato, Yukio Kawahara.

**Project administration:** Taisuke Nakahama, Yuki Kato, Yukio Kawahara.

**Resources:** Taisuke Nakahama, Yuki Kato, Yukio Kawahara.

**Software:** Yuki Kato.

**Supervision:** Taisuke Nakahama, Toshiharu Shibuya, Yuki Kato, Yukio Kawahara.

**Validation:** Jung In Kim, Taisuke Nakahama, Ryuichiro Yamasaki, Pedro Henrique Costa Cruz, Tuangtong Vongpipatana, Maal Inoue, Nao Kanou, Yanfang Xing, Hiroyuki Todo, Toshiharu Shibuya, Yuki Kato, Yukio Kawahara.

**Visualization:** Taisuke Nakahama, Ryuichiro Yamasaki, Yuki Kato, Yukio Kawahara.

**Writing – original draft:** Jung In Kim, Taisuke Nakahama, Yuki Kato, Yukio Kawahara.

**Writing – review & editing:** Jung In Kim, Taisuke Nakahama, Yuki Kato, Yukio Kawahara.

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
