## [Decision Letter · Decision Letter 0]

9 Feb 2021

Dear Dr Kawahara,

Thank you very much for submitting your Research Article entitled 'RNA editing at a limited number of sites is sufficient to prevent MDA5 activation in the mouse brain' to PLOS Genetics.

The manuscript was fully evaluated at the editorial level and by independent peer reviewers. The reviewers appreciated the attention to an important topic but identified some concerns that we ask you address in a revised manuscript

We therefore ask you to modify the manuscript according to the review recommendations. Your revisions should address the specific points made by each reviewer.

[LINK]

Yours sincerely,

Kent Hunter

Associate Editor

PLOS Genetics

Gregory Barsh

Editor-in-Chief

PLOS Genetics

Reviewer's Responses to Questions

**Comments to the Authors:**

Reviewer #1: In this study by Kim et al., the authors generate different ADAR1 p110 knockout mice and assess A->G editing in the transcripts. As there is very little expression of p150 isoform in the brain, the authors propose that p110 may be involved in the editing of endogenous dsRNA and preventing MDA5 activation. First, the author generate a p110 specific KO mice by mutating the start codon M294. Unfortunately, this resulted in the expression of a truncated form of p110, likely due to translation initiation from an internal AUG. Next, the authors generate another version of p110 KO mice by deleting the constitutive ADAR1 promoter along with Exon1B and 1C, and successfully derive mice lacking p110; however, the majority of p110-/- mice die within 2 days after birth. Unlike ADAR1-/- or p150-/-, p110-/- mice do not exhibit increased expression of interferon related genes. This early lethality can be rescued by expressing editing mutant of ADAR1E861A but not by concurrent deletion of MDA5, suggesting that early death is unrelated to RNA editing functions of p110. Using this novel p110-/- mice, the authors investigated the alterations A->G mutations in the transcriptome and observed only partial reduction in A->G mutations. Through concurrent deletion of p110-/- and ADAR2-/-, the authors demonstrate that both p110 and ADAR2 are contributing to A->G mutations. The authors also identify a few remaining mutations that may be attributed to p150 but not conclusively. This is a novel and well executed study. The manuscript is also well written. I have some minor comments.

1. For p110-/- mice, the authors indicate that the contributions of p110 expressed from a cryptic promoter or from p150 mRNA through the internal M294 is minimal. The authors should treat p110-/- cells with IFN treatment and assess p110 expression, as previously suggested for human cells (Cell. 2018;172(4):811–24.e14.; https://doi.org/10.1371/journal.ppat.1003963;
https://doi.org/10.1371/journal.ppat.1008842)

2. The authors should also assess editing in ADAR1E861A/p110-/- mice to rule out any residual editing by E861A mutant.

3. It will be informative to the readers if the authors could include A->G editing in the repeat sequences (SINE). This could provide information to readers about the levels of p150 editing.

4. LN 39- p is missing in p150

Reviewer #2: The ADAR family of RNA editing enzymes are important for modulating the structure and coding properties of the mammalian transcriptome. Here the authors describe the generation and characterization of mice genetically modified to express only the p150 isoform of ADAR1 and not the p110 isoform of ADAR1 or ADAR2. Since these three proteins are the only editing-functional mammalian ADARs, these mice provide an important tool for the study of phenotypes associated with p150 and p150-specific editing sites. Other labs have shown that ADAR1 p150 activity is essential to block recognition by duplex RNA sensors (e.g. MDA5) of the duplex RNA structure present in the transcriptome. However, a critical question that has been dogging the editing community is “What is the identity of the critical RNAs modified by ADAR1 that would otherwise by recognized by dsRNA receptors and trigger and immune response?” This study provides key data that start to address this question. By removing the other functional ADARs from the mouse genome, these authors can identify the sites edited by ADAR1 p150. The results reported in Figure 9 described highly efficient ADAR1 p150 editing sites, particularly in the 3’ UTR of the Mad2ll transcript, are very important. The conclusions of this paper are supported well by the results presented. I have only the following minor comments.

1) The introduction is quite wordy with the last paragraph essentially providing a synopsis of the entire study. The authors may wish to move some of this text to the discussion.

2) The discussion of the truncated ADAR1 p110 found in the M249A/M249A mice is distracting and does not add to the overall impact of the paper. The authors may consider removing this entirely from the manuscript.

3) An obvious question for these authors to address is whether edited Mad2ll 3’-UTR interacts with dsRNA sensor proteins like MDA5 differently than the unedited RNA. While I don’t believe these results are essential for publication, they could substantially raise the impact of the current work.

Reviewer #3: Review is uploaded

**Have all data underlying the figures and results presented in the manuscript been provided?**

Reviewer #1: Yes

Reviewer #2: Yes

Reviewer #3: Yes

PLOS authors have the option to publish the peer review history of their article (what does this mean?). If published, this will include your full peer review and any attached files.

Reviewer #1: No

Reviewer #2: No

Reviewer #3: **Yes: **Alan Herbert

---

## [Decision Letter · Decision Letter 1]

28 Mar 2021

Dear Dr Kawahara,

We are pleased to inform you that your manuscript entitled "RNA editing at a limited number of sites is sufficient to prevent MDA5 activation in the mouse brain" has been editorially accepted for publication in PLOS Genetics. Congratulations!

Yours sincerely,

Kent Hunter

Associate Editor

PLOS Genetics

Gregory Barsh

Editor-in-Chief

PLOS Genetics

Comments from the reviewers (if applicable):

Reviewer's Responses to Questions

**Comments to the Authors:**

Reviewer #1: The authors have addressed all of the previous concerns well.

Reviewer #3: I believe the authors have addressed the reviewer concerns. Nice job!

**Have all data underlying the figures and results presented in the manuscript been provided?**

Reviewer #1: None

Reviewer #3: Yes

PLOS authors have the option to publish the peer review history of their article (what does this mean?). If published, this will include your full peer review and any attached files.

Reviewer #1: No

Reviewer #3: **Yes: **Alan Herbert

**Data Deposition**

http://datadryad.org/submit?journalID=pgenetics&manu=PGENETICS-D-20-01959R1

**Press Queries**

---

## [Editor Report · Acceptance letter]

14 Apr 2021

PGENETICS-D-20-01959R1 

RNA editing at a limited number of sites is sufficient to prevent MDA5 activation in the mouse brain 

Dear Dr Kawahara, 

We are pleased to inform you that your manuscript entitled "RNA editing at a limited number of sites is sufficient to prevent MDA5 activation in the mouse brain" has been formally accepted for publication in PLOS Genetics! Your manuscript is now with our production department and you will be notified of the publication date in due course.

With kind regards,

Alice Ellingham

PLOS Genetics

On behalf of:
